# On Training Large Language Models for Long-Horizon Tasks: An Empirical Study of Horizon Length

**Sunghwan Kim** [1 2 †]  **Junhee Cho** [1]  **Beong-woo Kwak** [1 2 †]  **Taeyoon Kwon** [1]  **Liang Wang** [2]  **Nan Yang** [2]
**Xingxing Zhang** [2]  **Furu Wei** [2]  **Jinyoung Yeo** [1]

## Abstract

Large language models (LLMs) have shown promise as interactive agents that solve tasks through extended sequences of environment interactions. While prior work has primarily focused on system-level optimizations or algorithmic improvements, the role of task *horizon length* in shaping training dynamics remains poorly understood. In this work, we present a systematic empirical study that examines horizon length through controlled task constructions. Specifically, we construct controlled tasks in which agents face identical decision rules and reasoning structures, but differ only in the length of action sequences required for successful completion. Our results reveal that increasing horizon length alone constitutes a training bottleneck, inducing severe training instability driven by exploration difficulties and credit assignment challenges. We demonstrate that horizon reduction is a key principle to address this limitation, stabilizing training and achieving better performance in long-horizon tasks. Moreover, we find that horizon reduction is related to stronger generalization across horizon lengths: models trained under reduced horizons generalize more effectively to longer-horizon variants at inference time, a phenomenon we refer to as horizon generalization.

## 1. Introduction

Large language models (LLMs) are increasingly deployed as interactive agents that solve real-world tasks through interaction with environment. Coding agents such as Claude Code ([Anthropic, 2025c](#)) and Codex ([OpenAI, 2025](#)) illustrate this shift, performing multi-step workflows for iterative code debugging and decision-making.

Existing literature primarily explores two directions for enhancing these capabilities. The first focuses on system-centric optimizations, such as context engineering ([Anthropic, 2025b](#); [Zhang et al., 2025a](#); [Liu et al., 2025b](#)) and workflow orchestration ([Zhang et al., 2024](#); [Niu et al., 2025](#); [Zhang et al., 2025b](#)). The second targets model-centric optimization via Supervised Fine-Tuning (SFT) ([Liu et al., 2024](#); [Prabhakar et al., 2025](#); [Liu et al., 2025c](#)) and Reinforcement Learning (RL) ([Jin et al., 2025](#); [Lu et al., 2025b](#);[a](#)). Despite these advancements, both lines of research largely remain incremental extensions of single-turn paradigms, overlooking the fundamental challenges introduced by horizon length.

As the horizon increases, long-horizon tasks theoretically require higher step accuracy to succeed ([Sinha et al., 2025](#)). Simultaneously, the combinatorial growth in state-action mapping complexity makes exploration exponentially more difficult ([Park et al., 2025](#)). In addition, delayed feedback over long interactions makes credit assignment ambiguous, requiring a single return signal to be propagated across many steps and resulting in high-variance gradient estimates and noisy learning signals. Despite these challenges, horizon itself remains underexplored as a primary factor shaping training dynamics. Recent work ([Shen et al., 2025](#); [Xi et al., 2025a](#); [Bai et al., 2026](#)) has begun to address this gap using horizon-based curricula. However, these studies predominantly view horizon as an environmental constraint (i.e., interaction budget) rather than an intrinsic task property. Consequently, it remains unclear: *how does the horizon required to solve a task influence the training of LLMs?*

In this work, we conduct a systematic empirical study to investigate the effect of horizon length on training LLMs. To rigorously define our scope, we formalize the concept of "horizon" as illustrated in Figure [1](#). Based on this definition, we construct controlled task suites that decouple goal distance from reasoning complexity. Under these controlled experiments, we observe that the policies that learn reliably on short goal distances often exhibit severe instability as the goal distance increases, frequently leading to

---

[†]Work done during internship at Microsoft Research.
[1]Department of Artificial Intelligence, Yonsei University
[2]Microsoft Research. Correspondence to: Sunghwan Kim <kimsh8564@yonsei.ac.kr>.

*Proceedings of the 43 rd International Conference on Machine Learning*, Seoul, South Korea. PMLR 306, 2026. Copyright 2026 by the author(s).

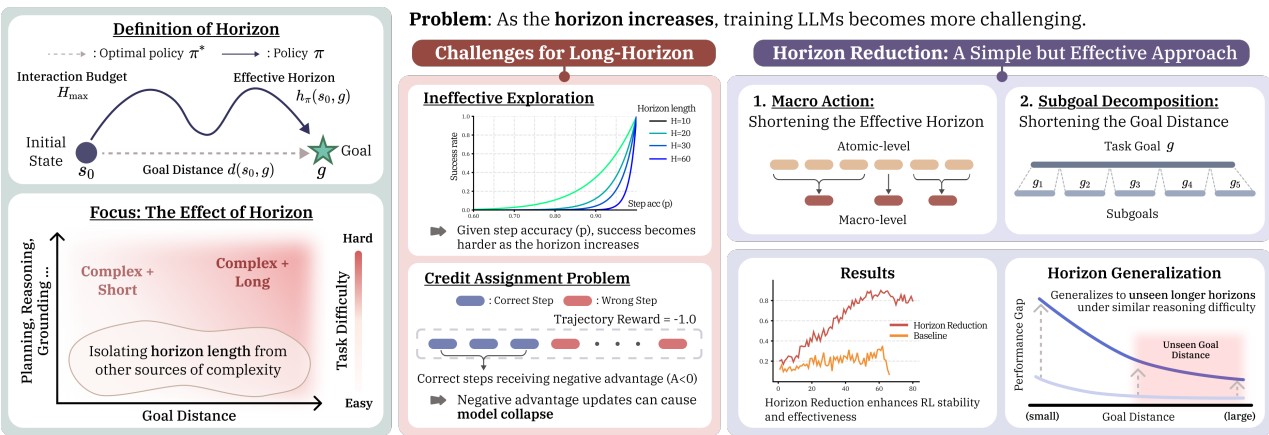

*Figure 1.* **A summary of our contributions.** In this work, we study the training of long-horizon LLM agents from a horizon-centric perspective and identify horizon length as a fundamental bottleneck. We show that horizon reduction stabilizes RL and strengthens the tendency toward horizon generalization on longer tasks with similar reasoning difficulty.

performance collapse. Crucially, this degradation occurs even when the underlying reasoning complexity remains unchanged, demonstrating that horizon length acts as an independent factor inducing fundamental training instability.

Beyond diagnosing these limitations, we investigate how they can be addressed. We identify *horizon reduction* as a simple yet powerful principle for training LLMs on long-horizon tasks. These approaches significantly improve both training stability and final performance. Moreover, we find that trained models can generalize to previously unseen longer horizons at inference time, a phenomenon we term *horizon generalization*. Overall, our results highlight horizon as a central factor shaping the training dynamics and suggest a scalable path toward robust long-horizon behavior without resorting to complex training procedures.

## 2. Preliminaries

### 2.1. From LLMs to LLM Agents

**Autoregressive language models.** We model a large language model (LLM) parameterized by $\theta$ as a stochastic policy $\pi_\theta$ in a token-level Markov decision process (MDP). At each generation step $i$, the state $s_i$ consists of the given context (i.e., prompt) $x$ and the sequence of previously generated tokens $(y_1, \ldots, y_{<i})$. The policy $\pi_\theta(\cdot \mid x, y_{<i})$ defines a categorical distribution over the vocabulary $\mathcal{V}$, from which the next token $y_i$ is sampled. This action induces a deterministic transition to the next state $s_{i+1} = (x, y_{\leq i})$. For a generated sequence $y = (y_1, \ldots, y_L)$ conditioned on $x$, its likelihood under $\pi_\theta$ factorizes as $\pi_\theta(y \mid x) = \prod_{i=1}^{L} \pi_\theta(y_i \mid x, y_{<i})$, where $L = |y|$ denotes the sequence length.

**LLM agents.** We model an LLM as a stochastic policy $\pi_\theta$ interacting with an environment over a finite horizon of $T$ steps. At each time step $t$, the agent generates an action $a_t \sim \pi_\theta(\cdot \mid s_t)$ conditioned on the current state $s_t$. The state is defined as $s_t = x_t = (g, m_t, o_t)$, where $x_t$ as current context, $g$ denotes the task goal, $m_t$ represents the agent's memory, and $o_t$ is the current observation from the environment. The interaction between the agent and the environment generates a trajectory $\tau = (s_0, a_0, s_1, \ldots, s_{T-1}, a_{T-1}, s_T)$, which captures the sequential decision-making process.

### 2.2. Training LLMs on Long-Horizon Tasks

#### 2.2.1. SUPERVISED FINE-TUNING

Supervised Fine-Tuning (SFT) is one of the most widely used paradigms for training LLM agents via token-level behavior cloning, maximizing the likelihood of expert trajectories (Liu et al., 2024; Prabhakar et al., 2025; Fang et al., 2025). Given demonstrations collected from expert policies, SFT trains the model to imitate action sequences conditioned on the interaction history, enabling the agent to implicitly acquire environment dynamics. This makes SFT both a strong initialization strategy and a reliable method for training LLM agents (Vattikonda et al., 2025).

#### 2.2.2. REINFORCEMENT LEARNING

**Background.** Reinforcement learning (RL) has a long history of success across a wide range of decision-making (Silver et al., 2017; Vinyals et al., 2019; Berner et al., 2019), yet its large-scale adoption for training LLMs is a relatively recent development (Ouyang et al., 2022; Jaech et al., 2024). Initial efforts relied on PPO (Schulman et al., 2017), but the requirement for a separate value network creates significant scalability bottlenecks. The emergence of critic-free

policy optimization methods, Group Relative Policy Optimization (GRPO) (Shao et al., 2024), has fundamentally shifted this paradigm. This substantially reduces memory and computational overhead while maintaining strong empirical performance, catalyzing variants including DAPO (Yu et al., 2025), GSPO (Zheng et al., 2025), CISPO (Chen et al., 2025a), among others (Liu et al., 2025d; Gao et al., 2025).

In classical deep RL, long-horizon challenges are typically addressed using value-based variance reduction. However, recent studies (Wang et al., 2025a; Park et al., 2025) suggest that the efficacy of these methods degrades as the horizon length increases. Inspired by the recent success of critic-free methods (Guo et al., 2025; Yang et al., 2025; Team et al., 2025), we pursue an alternative direction: *applying critic-free RL methods to long-horizon tasks*. This perspective not only simplifies the training pipeline but also opens new opportunities for scalable and stable RL in LLMs.

**Back to Basics: REINFORCE.** In this work, we revisit the fundamental on-policy algorithm, REINFORCE (Williams, 1992). The core objective of this method is to maximize the expected cumulative return $G_t = \sum_{k=t}^{T-1} \gamma^{k-t} r_k$, which represents the discounted sum of rewards from time step $t$. To reduce the high variance associated with raw returns while maintaining unbiasedness, it is common practice to subtract $b_t$ a function independent of the action. Defining the advantage function as $A_t := G_t - b_t$, the policy gradient objective is formally expressed as

$$\nabla \mathcal{J}(\theta) = \mathbb{E}_{\tau \sim \pi_\theta} \left[ \sum_t A_t \nabla_\theta \log \pi_\theta(a_t \mid s_t) \right].$$

**Reward design.** To decouple goal achievement from local constraint satisfaction, we separate the reward signal into a trajectory-level and a step-level component, ensuring that penalties associated with individual steps do not contaminate the learning signal for task success.

- $r_t^{\text{traj}}$: Computed as the discounted return $\sum_{k=t}^{T-1} \gamma^{k-t} r_k$.

- $r_t^{\text{step}}$: Defined as $r_t^{\text{format}} + r_t^{\text{valid}}$, where components penalize parsing errors and invalid actions, respectively.

To stabilize optimization, we apply batch normalization to each component (i.e., $\hat{r}_t^{\text{traj}}$ and $\hat{r}_t^{\text{step}}$): $\hat{r}_k = (r_k - \text{mean}(\{r_k\}_{k=1}^B)) / \text{std}(\{r_k\}_{k=1}^B)$, where $B$ is batch size. The final advantage estimate is then constructed as: $A_t = \hat{r}_t^{\text{traj}} + \alpha \cdot \hat{r}_t^{\text{step}}$, with $\alpha$ controlling the relative weight of the step-level reward. We set $\alpha = 0.2$ in all experiments.

**Stabilizing off-policy REINFORCE.** Strict on-policy optimization is often impractical in LLM pipelines due to computational constraints. Trajectories are typically reused

across updates (i.e., mini-batch update), and the architectural decoupling of inference (e.g., vLLM (Kwon et al., 2023)) and training (e.g., FSDP (Zhao et al., 2023)) inevitably introduces policy staleness where the sampling policy $\mu_{\theta_{\text{old}}}$ diverges from the current $\pi_\theta$. To mitigate the bias arising from this distribution shift, we optimize the policy by maximizing the following weighted objective:

$$\nabla \mathcal{J}(\theta) = \mathbb{E}_{\tau \sim \mu_{\theta_{\text{old}}}} \left[ \sum_{t=0}^{T-1} w_t A_t \sum_{i=1}^{|y_t|} \nabla \log \pi_\theta(y_{t,i} | x_t, y_{t,<i}) \right]$$

Here, $w_t$ is an importance sampling weighted term designed to address distribution shift and treated as a constant coefficient (i.e., stop gradient). It combines Masked Importance Sampling (MIS) based on the geometric mean ratio with Truncated Importance Sampling (TIS) based on the sequence-level ratio (Yao et al., 2025; Liu et al., 2025a):

$$w_t = \underbrace{\mathbb{I}(C_{\text{low}} \leq \rho_{\text{geo},t} \leq C_{\text{high}})}_{\text{Masked IS}} \cdot \underbrace{\min(\rho_{\text{seq},t}, C)}_{\text{Truncated IS}},$$

where $\rho$ denotes the IS ratio. Details of the RL design and hyperparameter are provided in Appendix C.3 and D.3.

**Token-level gradient dynamics.** To understand the roles of positive and negative advantage, Gao et al. (2025) analyze how gradients propagate through the logits $z$. For a sampled token $y_i$ with advantage $A_i$, the gradient of the objective with respect to the logit $z_v$ of any token $v \in \mathcal{V}$ is given by:

$$\nabla_{z_v} \mathcal{J}(\theta) = \begin{cases} (1 - \pi_\theta(y_i \mid x, y_{<i})) \cdot A_i, & v = y_i \\ -\pi_\theta(v \mid x, y_{<i}) \cdot A_i, & v \neq y_i \end{cases}$$

This expression reveals a qualitative asymmetry between positive and negative advantages. When the advantage is positive, the gradient increases the logit of the sampled token while decreasing the logits of all other tokens, effectively concentrating probability mass on the chosen action. In contrast, when the advantage is negative, probability mass is removed from the sampled token and redistributed across all other tokens, resulting in a diffuse update over the vocabulary. Detailed derivations are provided in Appendix D.1.

## 3. Evaluation Setup

### 3.1. Problem Setting

#### 3.1.1. FORMULATION

The concept of "horizon" in a task is multi-dimensional, encompassing the intrinsic property of the task, the constraints of the environment, and the behavior of the agent. We introduce the following formalisms within an interaction defined by an initial state $s_0$ and a task goal $g$:

*Table 1.* **Statistics of the dataset.** We separate tasks into levels L1–L7 based on the $d(s_0, g)$. The blue columns (L1–L4) represent the horizon levels included in the training set. The red columns (L5–L7) denote tasks with extended horizons unseen during training, used to evaluate horizon length generalization.

|  | **L1** | **L2** | **L3** | **L4** | **L5** | **L6** | **L7** |
|---|---|---|---|---|---|---|---|
| $d(s_0, g)$ | 11-15 | 16-20 | 21-25 | 26-30 | 31-35 | 36-40 | 41-45 |
| $N_{\text{train}}$ | 640 | | 640 | | - | - | - |
| $N_{\text{test}}$ | 100 | 100 | 100 | 100 | 100 | 100 | 50 |

- **Goal Distance** $d(s_0, g)$**:** The minimum number of atomic actions required to reach goal under an optimal policy $\pi^*$.

- **Interaction Budget** $H_{\max}$**:** The maximum number of interaction steps allowed by the environment.

- **Effective Horizon** $h_\pi(s_0, g)$**:** The actual number of steps a policy $\pi$ takes to successfully reach the goal.

For any successful trajectory, the inequality $d(s_0, g) \leq h_\pi(s_0, g) \leq H_{\max}$ holds. Here, the gap between $h_\pi$ and $d$ reflects the inefficiency of the policy, while the constraint $H_{\max}$ defines the boundary of feasibility.

### 3.1.2. FOCUS: THE EFFECT OF HORIZON

Our goal is to understand *how horizon length affects the training dynamics of LLM agents*. A key challenge in studying long-horizon tasks is that horizon length is typically entangled with other sources of difficulty. As tasks require longer interactions, they often simultaneously demand more sophisticated planning, reasoning, or perceptual capabilities. For example, a Sudoku puzzle with more empty cells typically requires not only a longer execution horizon but also more complex solving techniques. This coupling makes it unclear whether training failures stem from horizon length itself or from these confounding factors. To isolate horizon effects, we need a principled way to construct tasks that vary in goal distance while holding other complexity constant.

**Isolating horizon from solving complexity.** A straightforward approach would be to filter out tasks that a model fails to solve, interpreting failure as a lack of problem-solving ability and success as evidence of sufficient capability. However, models may fail in such settings due to error accumulation, context drift, or instability over extended interactions, rather than a lack of the underlying reasoning capability required to solve the task (Sinha et al., 2025). As a result, model failure on long-horizon tasks alone does not reliably indicate insufficient problem-solving ability.

To address this issue, we adopt the following assumption: if a model possesses sufficient reasoning capability required to solve a task, it should be able to demonstrate this capability in a simplified, short-horizon setting. Based on

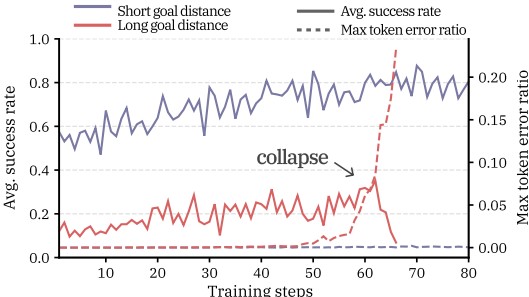

*Figure 2.* **Training dynamics on different goal distance.** While RL training is stable on short goal distance (L1–L2), it exhibits severe instability as the goal distance increases (L3–L4).

this assumption, we construct short-horizon proxy tasks by converting long-horizon tasks into equivalent single-step formulations (i.e., asking the model to generate a complete Sudoku solution in one step instead of solving it incrementally). Performance on these proxy tasks serves as a clean indicator of the model's solving capability, decoupled from the effects of horizon length.

We filter the task instances to retain only those that a given model can successfully solve in the short-horizon setting. Then, we partition the resulting instances into datasets according to their goal distance $d(s_0, g)$, enabling controlled analysis of training dynamics. The datasets constructed under this procedure are summarized in Table 1 and details of the dataset construction process are provided in Appendix B.

### 3.2. Task Environments

We use text-based games as our evaluation environments. Compared to GUI or tool environments, text-based games eliminate confounding factors such as visual grounding errors or domain-specific knowledge. More importantly, they allow precise control through procedural generation, making them well suited for studying horizon effects. We adopt Sudoku as our primary testbed to analyze training dynamics.

**Sudoku.** In Sudoku, the agent interacts with a $9 \times 9$ grid by taking actions to fill cells (e.g., `value(n, rXcY)`). This task is particularly well-suited for our analysis because modern LLMs possess substantial prior knowledge of Sudoku rules (see Appendix D.2). We control the horizon length by varying the number of empty cells, using this count as a direct proxy for the goal distance $d(s_0, g)$. To ensure that solving complexity remains constant across different horizon lengths, we verify our dataset using HoDoKu (Hobiger, 2008), a Sudoku solver that classifies puzzles by required techniques. Our dataset consists exclusively of puzzles solvable using basic techniques, ensuring that variations in solvability stem from horizon length. Detailed statistics for the training and test sets are provided in Table 1.

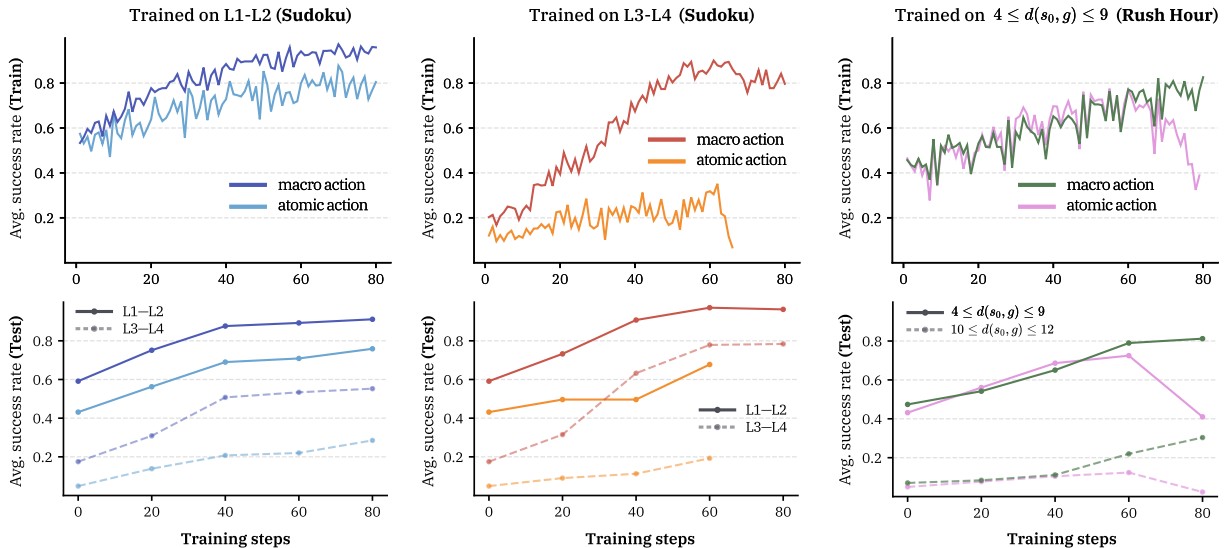

*Figure 3.* **Horizon reduction improves RL on long-horizon tasks.** Training and test success rate on Sudoku and Rush Hour with atomic actions versus macro actions across different goal distance regimes. Across both environments, using macro actions for horizon reduction leads to more stable and effective RL, particularly in a long goal distance setting.

**Rush Hour.** To verify that our insights are not confined to a single domain, we conduct validation experiments on Rush Hour. Rush Hour is a sliding-block puzzle where the agent must maneuver a target car to an exit using directional actions (e.g., move(id, direction)). Unlike Sudoku, this task emphasizes spatial reasoning and requires minimal domain-specific prior knowledge. Here, the minimum number of moves required for the optimal solution (min_moves) serves as the proxy for goal distance.

## 4. Training LLMs for Long-Horizon Tasks

### 4.1. Main Results

**Implementation details.** We employ Qwen3-1.7B (Yang et al., 2025) as our base model for all experiments. We first perform SFT on expert trajectories generated by larger models (e.g., GPT-5-mini), and then use the resulting model as the initial policy for RL, which is trained for 4 epochs. A temperature of 0.8 is used for both training rollout and inference. For evaluation, we sample 4 trajectories per instance to report pass$@K$ and avg$@K$. Comprehensive training details and hyperparameters are provided in Appendix C.

**RL on long-horizon tasks.** Starting from SFT-initialized policies, we apply RL across tasks with varying goal distances, comparing short goal distance (L1–L2) and longer (L3–L4) settings. As shown in Figure 2, we observe a pronounced divergence in training behavior as horizon length increases. While RL consistently improves performance on shorter goal distance tasks, training on L3–L4 instances leads to severe instability and catastrophic collapse. We further observe that training collapse is accompanied by a

sharp increase in the maximum-length response ratio, signaling a transition toward incoherent or excessively long generations. We conjecture that this phenomenon arises from the accumulation of erroneous negative-advantage updates, which progressively distort the policy and ultimately destabilize the generation process.

**Why RL fails on long-horizon tasks?** Training LLM agents with RL on long-horizon tasks introduces fundamental challenges that intensify as horizon length increases. First, *mapping complexity* grows non-linearly: as horizon length increases, the relationship between states and optimal actions becomes increasingly complex (Park et al., 2025). As the horizon increases, the state-action space explodes, and early decisions exert a disproportionate constraint on future outcomes. Consequently, the probability of adhering to an optimal trajectory decays exponentially, making the discovery of successful sequences difficult. Second, *credit assignment* becomes severely challenging under sparse rewards. When a trajectory fails, the entire sequence receives negative advantage, including intermediate steps that were individually correct. As discussed in Section 2.2.2, this negative feedback creates problematic gradient dynamics: to suppress sampled actions, the model diffuses probability mass across the entire vocabulary, inadvertently boosting irrelevant tokens while penalizing potentially optimal ones.

### 4.2. Horizon Reduction as a Key Principle

#### 4.2.1. METHOD

We observe that RL fails when the horizon length increases. While one might attempt to mitigate this by designing com-

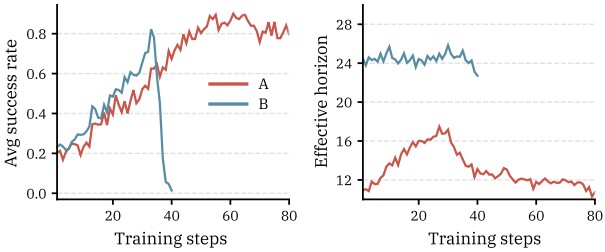

*Figure 4.* **RL stability depends on effective horizon.** We compare two settings with macro-action policy: **(A)** reduced effective horizon via macro actions, and **(B)** an artificially restored long-horizon setting by restricting execution to single atomic actions.

plex methods, we argue for a more fundamental solution.

> *"The best way to escape from a problem is to solve it."*

Instead of forcing the agent to learn intractable long-horizon dependencies, **horizon reduction** structurally minimizes the interaction length required to solve a task. Fundamentally, this approach aims to decrease the effective horizon $h_\pi(s_0, g)$ to a regime where RL remains stable and efficient. We identify two distinct mechanisms to achieve this: (a) Macro Actions and (b) Subgoal Decomposition.

**Macro actions.** Our definition of the goal distance $d(s_0, g)$ is based on atomic actions. By allowing the policy to operate over *macro actions*, which compose multiple atomic actions into higher-level primitives, we can naturally reduce the horizon length. Formally, a policy $\pi'$ defined over macro actions can achieve smaller effective horizon $h_{\pi'}(s_0, g)$ than the $h_\pi(s_0, g)$ of a policy restricted to atomic actions. For our experiments, we allow the agent to generate multiple actions per step in Sudoku, and multi cell moves (e.g., `move(id, direction, N)`) in Rush Hour.

**Subgoal decomposition.** Alternatively, we can decompose the global goal $g$ into a sequence of subgoals $(g_1, g_2, \ldots, g_k)$ rather than attempting to solve $g$ in a single episode. The total goal distance can then be expressed as the sum of the distances of these segments: $d(s_0, g) = \sum_{i=1}^{k} d(s_0^{(i-1)}, g_i)$, where $s_0^{(i-1)}$ denotes the initial state for the $i$-th subgoal. Since Sudoku possesses naturally verifiable subgoals (e.g., subgrid correctness), we focus our decomposition experiments on this domain.

#### 4.2.2. RESULTS

**A simple but effective approach.** We first examine whether reducing the effective horizon through macro actions can mitigate training collapse. As shown in Figure 3, incorporating macro actions yields substantial performance gains across all experimental settings. On short goal distance tasks (L1–L2), training with macro actions converges

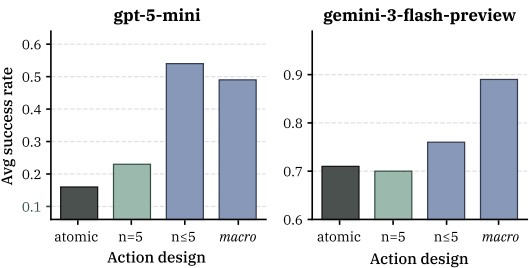

*Figure 5.* **Results of macro action design.** Flexible macro actions ($n \leq 5$ and *macro* ) show better performance than both atomic and fixed-length ($n = 5$) designs across models on Sudoku. Additional ablation results for $n$ are provided in Figure 13.

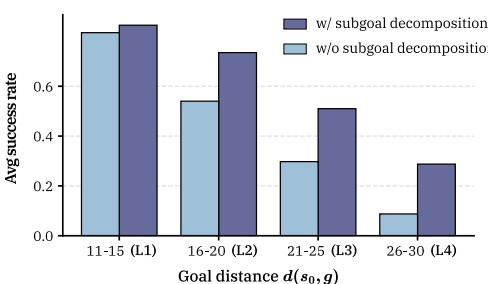

*Figure 6.* **Effect of subgoal decomposition on RL.** Average success rate on Sudoku across increasing goal distances.

faster and achieves higher final performance than the atomic-action baseline. More importantly, on long goal distance tasks (L3–L4), where training with atomic actions suffers catastrophic collapse, macro actions maintain stable learning and continue to improve performance. These results show that a simple structural modification to the action space can substantially improve both training stability and scalability in long-horizon tasks.

**Horizon as the critical bottleneck.** The observed benefits of macro actions may arise from two distinct factors: improved exploration enabled by a stronger base policy (initial performance in Figure 3), or a reduction in effective horizon. To isolate the specific contribution of horizon, we perform a controlled ablation in which we use the same macro-action policy but restrict it to execute only a single atomic step per turn. This intervention restores a long interaction horizon while preserving the policy's underlying representation. In the long horizon setting (**B** in Figure 4), performance initially improves but eventually collapses. In contrast, the horizon-reduced setting (**A** in Figure 4) exhibits slower yet converges to a high performance. These findings provide evidence that effective horizon $h_\pi(s_0, g)$ is a primary determinant of training stability.

**Design of macro action.** To examine macro actions on frontier models (GPT-5-mini, Gemini-3-Flash), we compare

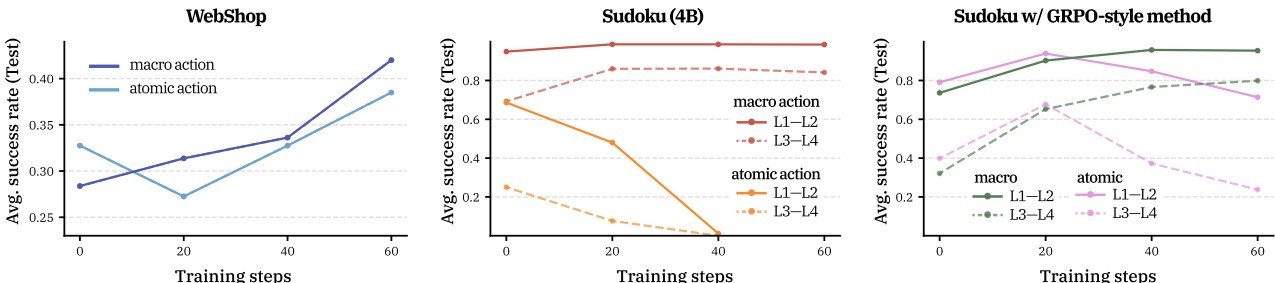

*Figure 7.* **Robustness of horizon reduction across diverse settings.** (*Left*) On WebShop, horizon reduction improves both training stability and average success rate. (*Middle*) On Sudoku (L3–L4) with a 4B model, training collapse persists under the default horizon, while horizon reduction yields stable improvement. (*Right*) Under a GRPO-style optimizer, the same instability pattern emerges and is resolved by horizon reduction. Across all three settings, horizon reduction consistently prevents collapse and improves final performance. Additional experimental details and training dynamics are provided in Appendix C.4.

three designs: (1) *atomic action*, (2) *fixed-length macro* (exactly $k$ steps), and (3) *flexible macro* (dynamic length, either bounded by $k$ or unbounded). Figure 5 shows that fixed-length macro actions perform worse due to rigidity and overshooting. In contrast, flexible macro consistently achieve the highest performance. This indicates that effective horizon reduction requires policy-controlled granularity: rigid constraints hurt performance, whereas policy-driven flexibility in action length is essential for robustness.

**Results of subgoal decomposition.** Another effective strategy for horizon reduction is *subgoal decomposition*. For this experiment, we construct a dense reward variant of Sudoku in which the agent receives intermediate rewards for correctly completing individual subgrids. We segment the interaction trajectory upon subgrid completion and compute the return $G_t$ independently for each segment. This design effectively breaks the original long-horizon objective into a sequence of shorter, verifiable subtasks. We train the resulting training on long goal distance regime (L3–L4) with subgoal decomposition, where the standard sparse-reward RL baseline previously fails to learn. To ensure a fair comparison, we match the training duration to the pre-collapse phase of the baseline, isolating the effect of reward structure from that of training time. As illustrated in Figure 6, we observe a substantial performance gap: while the sparse-reward baseline makes little progress, the subgoal-guided policy learns stably and achieves strong performance. These results underscore the effectiveness of subgoal decomposition for horizon reduction, and support the broader utility of process reward, which we discuss further in Section 5.

### 4.3. Robustness Across Environments, Model Scales, and Optimizers

Our main experiments rely on controlled environments to cleanly isolate horizon length from other factors such as reasoning difficulty, perception, and stochasticity. One might ask whether this instability is particular to puzzle domains, the 1.7B model scale, or our optimizer. We run three additional experiments to probe each of these dimensions.

**Environment.** To assess whether our findings generalize to more complex settings, we evaluate on WebShop (Yao et al., 2022a), a web-interaction benchmark involving natural-language observations, multi-step decision-making and partially observable. However, Figure 7 shows that reducing the horizon consistently improves both training stability and average success rate, suggesting that the benefits of horizon reduction extend to more realistic environments.

**Model scale.** Our main experiments use a Qwen3-1.7B. To examine whether increasing model capacity alleviates the horizon bottleneck, we repeat the Sudoku L3–L4 experiments with a 4B model. Under the atomic action setting, training still collapses at the larger scale, indicating that increased capacity alone does not resolve the issue. In contrast, applying horizon reduction enables the 4B model to avoid collapse and achieve higher performance. These results show that the horizon bottleneck persists across model scales, while horizon reduction remains an effective mitigation strategy irrespective of model capacity.

**Optimizer.** To verify that our findings are not specific to a particular optimization algorithm, we repeat the horizon comparison using a GRPO-style method with group-normalized advantages. Under the default horizon setting, performance initially improves but subsequently degrades over the course of training. In contrast, the horizon-reduced setting continues to improve steadily. This pattern closely mirrors the behavior observed in our default setup, indicating that the instability is not tied to a specific optimizer.

Taken together, these results support the view that effective horizon length is a cross-cutting bottleneck in long-horizon LLM agent training.

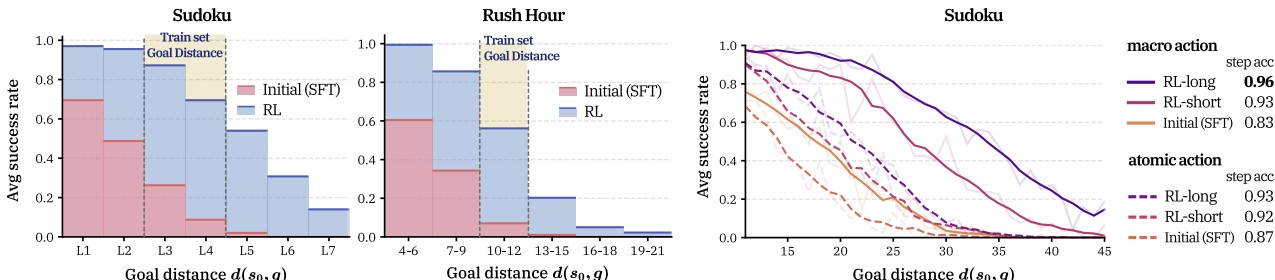

*Figure 8.* **Horizon generalization.** (*Left* and *middle*) Results on Sudoku and Rush Hour demonstrate that policies trained on limited goal distance ranges generalize effectively to unseen horizons. (*Right*) Success rates on Sudoku as a function of goal distance for models with different step accuracy reveal that macro-action policies consistently outperform atomic actions across horizons. RL-short and RL-long are trained on L1-L2 and L3-L4, respectively. See Appendix D.4 for step accuracy details and Rush Hour results.

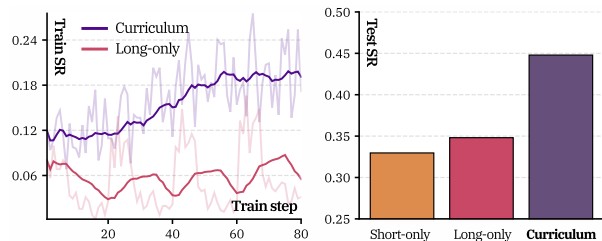

*Figure 9.* **Horizon curriculum.** On Rush Hour, we compare three training strategies: *Short-only* trains on $4 \leq d(s_0, g) \leq 9$, *Long-only* trains on $10 \leq d(s_0, g) \leq 12$, and **Curriculum** first trains on short horizons then continues on long horizons.

### 4.4. In-Depth Analysis

**Horizon generalization.** We evaluate whether models trained on a fixed range of goal distances can generalize to unseen horizons. As shown in Figure 8, models trained on moderate goal distance not only perform well on shorter tasks, but also achieve substantial gains on longer horizons. Moreover, the performance gap between our method and the baseline grows as goal distance increases, a phenomenon we refer to as *horizon generalization*.

Figure 8 (right) provides further insight into the mechanisms underlying this effect. Models trained with horizon reduction (e.g., macro action) show consistently stronger horizon generalization trends, which are partly explained by improved per-step accuracy. By stabilizing training and simplifying decision-making, horizon reduction leads to higher step accuracy, which is especially important in long-horizon settings where performance is highly sensitive to per-step errors. As a result, these models maintain strong performance even on unseen, longer horizons. In addition, Figure 8 (right) also reveals cases in which horizon-reduced models with lower per-step accuracy outperform atomic-action baselines with higher step accuracy. By reducing the effective horizon, macro actions decrease the number of decision points required to reach the goal, creating fewer opportunities for errors to accumulate.

**Curriculum learning via horizon generalization.** As shown in Figure 9, training directly on tasks with goal distances between 10 and 12 yields minimal improvement, likely due to insufficient initial performance to sustain optimization. Hypothesizing that we can bootstrap long-horizon capability via horizon generalization, we implement a curriculum strategy: we first train a policy on shorter distances ($4 \leq d(s_0, g) \leq 9$) and use it to initialize training for the longer target tasks ($10 \leq d(s_0, g) \leq 12$). This results in marked performance gains, confirming that establishing competence on shorter horizons is a prerequisite for learning at longer horizons. While this result aligns with prior work employing curricula based on maximum step limits ($H_{\max}$) (Shen et al., 2025; Xi et al., 2025a), we uniquely frame the progression in terms of intrinsic goal distance.

## 5. Discussion

Our findings suggest a shift in how long-horizon LLM agents should be designed and trained, highlighting horizon length as a fundamental bottleneck and motivating future work on horizon-aware training and system design.

**Horizon reduction via action abstraction.** A natural strategy for horizon reduction is to allow agents to operate over higher-level actions that encapsulate multiple atomic actions within a single step. We argue that the success of such abstractions stems not just from their representational expressiveness, but from their ability to drastically shorten the effective interaction horizon. Code-based agents exemplify this: by generating executable programs with loops and conditionals, they collapse long sequences of tool invocations into compact executions, directly mitigating error accumulation and credit assignment instability (Wang et al., 2024a; Anthropic, 2025a). A similar dynamic governs GUI-based agents, where augmenting low-level clicks with high-level API calls reduces the decision count and stabilizes learning (Song et al., 2025). While these systems often adopt abstraction for efficiency, they can be understood as

implicitly applying horizon reduction. Explicitly prioritizing such action space designs offers a systematic path to overcoming the optimization barriers of long-horizon tasks.

**Horizon reduction via subgoal decomposition.** A complementary mechanism for horizon reduction is subgoal decomposition, which restructures a long-horizon objective into a sequence of shorter, tractable segments. This approach aligns with hierarchical reinforcement learning, where a high-level policy transforms a long-horizon problem into a series of short-horizon subproblems (Zhou & Zanette, 2024; Hu et al., 2025). Recent LLM-based agents similarly employ planners or explicit milestone tracking to manage extended tasks (Erdogan et al., 2025; Chae et al., 2025). From the lens of our analysis, the primary benefit of these methods lies in their ability to localize credit assignment and constrain optimization to short effective horizons. Integrating process reward models with verifiable subgoals further amplifies this effect by providing dense, intermediate feedback, thereby reducing gradient variance and preventing the collapse modes we observe in direct long-horizon optimization (Xi et al., 2025b; Lee et al., 2025).

While much of the field focuses on increasingly complex RL algorithms or domain-specific methods, we argue that horizon-aware design should come first. Our results provide both empirical and conceptual evidence that designing horizon reduction environment and leveraging horizon generalization constitutes a fundamental strategy for building capable long-horizon LLM agents.

## 6. Related Work

**LLMs for long-horizon tasks.** LLMs have evolved from simple question-answering systems (Wang et al., 2022; Wei et al., 2022; Asai et al., 2024) into interactive agents capable of executing extended sequences of actions to solve complex problems (Yao et al., 2022b; Wang et al., 2024b). These long-horizon capabilities are essential across a wide range of applications, including software engineering (Jimenez et al., 2023; Xu et al., 2024), web automation (Zhou et al., 2023; Chae et al., 2025), and embodied control (Kwon et al., 2025). Recent progress has been driven primarily by inference-time innovations rather than advances in training methodologies. By leveraging frontier models, research has focused on architectural strategies such as *context engineering* (Anthropic, 2025b; Zhang et al., 2025a; Liu et al., 2025b) and *structured workflow* (Zhang et al., 2024; Niu et al., 2025; Zhang et al., 2025b), which decomposes complex goals into hierarchical plans or iterative refinement loops.

**Training LLM agents.** Early approaches primarily relied on SFT utilizing expert and synthetic demonstrations to establish agent capabilities (Liu et al., 2024; Prabhakar

et al., 2025; Liu et al., 2025c). More recently, studies (Jin et al., 2025; Chen et al., 2025b) have increasingly explored RL-based training paradigms that enable agents to learn through interaction. For example, online filtered behavioral cloning (Bai et al., 2024) combines on-policy exploration with selective imitation to reduce variance in policy updates, while other approaches focus on improving value estimation by pretraining critics prior (Chen et al., 2025c; Wang et al., 2025b). More recently, critic-free methods have emerged as a scalable alternative (Lu et al., 2025a; Liu et al., 2025e). Notably, group-in-group policy optimization (GiGPO) (Feng et al., 2025) has further advanced step-level credit assignment in multi-turn tasks through hierarchical grouping of trajectories and states. However, these methods may be less effective in environments with complex, high-dimensional state spaces. Beyond post-training methods, Su et al. (2025) introduce continual pretraining that emphasizes modeling action consequences and environment dynamics.

## 7. Conclusion

In this work, we identify horizon length as a fundamental bottleneck in training LLM agents, independent of intrinsic reasoning complexity. Through systematic experiments, we demonstrate that increasing horizon length alone induces severe training instability, primarily driven by intractable exploration and noisy credit assignment. To overcome these limitations, we establish horizon reduction as a critical design principle. We propose horizon reduction as a key training principle and uncover horizon generalization, where RL-trained models successfully solve unseen horizon tasks. Ultimately, we argue that managing the effective horizon is a fundamental prerequisite for scalable learning, suggesting that horizon-aware design must precede algorithmic sophistication in the development of long-horizon agents.

## Impact Statement

This work aims to advance the understanding of how horizon length affects the training dynamics of LLMs. Our contributions are primarily methodological, focusing on controlled empirical analysis and training principles for improving stability and generalization in long-horizon RL. The techniques studied in this paper may support the development of more reliable and capable agentic systems, which could have downstream benefits in domains such as software assistance, scientific workflows, and automation. At the same time, improved long-horizon agents could potentially amplify risks associated with misuse of autonomous systems, including unintended behavior in complex environments. These risks are not unique to our work and are broadly studied in the literature on AI safety and alignment. We do not introduce new data sources, user-facing applications, or deployment-specific mechanisms, and our experiments are

limited to synthetic, controlled environments. We therefore do not foresee immediate negative societal impacts arising directly from this work.

# Acknowledgements

We thank Minju Kim, Namyoung Kim, Minseok Kang, Yeonjun Hwang, Hyojun Kim, Dongjin Kang, and the anonymous reviewers for their valuable discussions and feedback. This project is supported by Microsoft Research Asia. This research was supported by the MSIT(Ministry of Science, ICT), Korea, under the Global Research Support Program in the Digital Field program(RS-2024-00436680, 70%) and ITRC(Information Technology Research Center) support program(IITP-2026-RS-2020-II201789, 30%) supervised by the IITP(Institute for Information & Communications Technology Planning & Evaluation).

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

# A. Limitations

While our work provides systematic evidence for the impact of horizon length on training LLM agents, several limitations warrant discussion.

**Task domains.**    Our experiments focus on text-based games, such as Sudoku and Rush Hour, which offer controlled settings for isolating horizon effects. Real-world agent deployments, including web navigation, software engineering workflows, and robotic manipulation, introduce additional challenges beyond horizon length, such as visual perception, noisy observations, and stochastic environment dynamics. Although we expect the core difficulties induced by long horizons to persist in these settings, validating our findings across more realistic and heterogeneous domains remains an important direction for future work.

**Model scale and diversity.**    In this work, we focus on relatively small models (i.e., Qwen3-1.7B) to enable controlled analysis of horizon effects. Larger models may exhibit different behaviors: increased capacity could mitigate certain horizon-induced instabilities through improved memorization or pattern recognition, or alternatively introduce new failure modes that arise only at scale. Due to resource constraints, we do not evaluate frontier-scale models in this study. In addition, our experiments are conducted using a single model family, Qwen. While models of comparable size but different architectures or training corpora may exhibit distinct learning dynamics, we believe that the impact of horizon length represents a fundamental challenge that is largely orthogonal to specific model choices. Understanding how horizon effects interact with model scale and diversity remains an important direction for future work.

Despite these limitations, we believe our controlled experimental approach provides foundational insights into horizon-induced training difficulties and offers a useful experimental framework for studying long-horizon behavior across diverse tasks, model scales, and training paradigms.

# B. Dataset Construction

### B.1. Sudoku Dataset

We construct our Sudoku evaluation datasets following a three-stage procedure designed to isolate horizon effects from solving complexity while ensuring sufficient data diversity.

**Stage 1: Candidate puzzle collection.**    We assemble an initial pool of candidate puzzles from two sources: Ritvik19's Sudoku Dataset[1] and additional puzzles generated using HoDoKu (Hobiger, 2008), a Sudoku generator and solver that allows control over puzzle difficulty through technique classification.

**Stage 2: Isolating solving complexity.**    As discussed in Section 3.1.2, we filter puzzles to retain only those solvable in a short-horizon proxy task. We convert each puzzle into a single-turn formulation where the model must generate the complete solution board in one response, testing latent reasoning capability without multi-step interaction effects. Using Qwen3-8B (Yang et al., 2025) with pass@8 sampling (temperature 0.7), we keep only puzzles with at least one correct solution across eight attempts.

**Stage 3: Partitioning by goal distance.**    We partition the filtered puzzles into seven datasets (L1–L7) based on the number of empty cells, which serves as a proxy for goal distance $d(s_0, g)$. For L1–L4, we randomly split puzzles into train and test sets, yielding 640 training tasks and 100 test tasks for each of L1–L2 and L3–L4. L5–L7 are used solely for evaluation, with 100 test tasks each for L5–L6 and 50 for L7. Detailed statistics are reported in Table 1.

### B.2. Rush Hour Dataset

We apply the same three-stage procedure to Rush Hour with the following domain-specific adaptations.

**Stage 1: Candidate puzzle collection.**    We generate candidate puzzles by randomly placing vehicles on a fixed 6×6 board, varying their count, positions, orientations, and sizes. We then filter for valid puzzles using Fogleman's Rush Hour solver[2],

---

[1]https://huggingface.co/datasets/Ritvik19/Sudoku-Dataset

[2]https://github.com/fogleman/rush

which verifies solvability and computes the minimum number of moves required.

**Stage 2: Isolating solving complexity.** Unlike Sudoku, converting Rush Hour into a pure single-turn formulation resulted in poor model performance, as the task inherently requires observing board states after each move to inform subsequent decisions. Therefore, we adopt a multi-turn setting but minimize horizon length by allowing the model to move vehicles multiple cells at once in a single action. We evaluate using GPT-5-mini with pass@1 sampling and retain only puzzles the model can solve under this compressed-horizon setting.

**Stage 3: Partitioning by goal distance.** We partition puzzles based on `min_moves`—the theoretical minimum number of moves in an optimal solution computed by Fogleman's solver—which serves as a proxy for goal distance $d(s_0, g)$. We sample 100 puzzles for each of six goal distance ranges, as detailed in Table 2.

*Table 2.* Rush Hour dataset statistics partitioned by goal distance.

| $d(s_0, g)$ | 4–6 | 7–9 | 10–12 | 13–15 | 16–18 | 19–21 |
|---|---|---|---|---|---|---|
| $N_{\text{test}}$ | 100 | 100 | 100 | 100 | 100 | 100 |

### B.3. SFT dataset.

For training LLMs with SFT, we collect expert demonstrations using high-performance open-weights and proprietary models. For Sudoku, initial trajectories are sampled from Qwen3-32B and filtered based on correctness. To address the issue of redundant reasoning steps in the raw CoT, we employ GPT-5-mini to distill these paths into more efficient reasoning chains. Conversely, for Rush Hour, we obtain successful trajectories directly from GPT-5-mini without additional refinement.

## C. Implementation Details

### C.1. LLM Agents

**Memory management.** Performance on long-horizon tasks is fundamentally constrained by the *self-conditioning effect* (Sinha et al., 2025), whereby errors accumulated from an agent's own past generations progressively degrade future predictions. In addition, LLM agents often struggle to effectively utilize long contexts, further compounding the difficulty of long-horizon execution (Yen et al., 2025; Anthropic, 2025b). To control this effect, we adopt a non-growing memory setting in which the agent does not retain the full interaction history. At each step $t$, the agent's context $m_t$ is restricted to a sliding window containing only the most recent $K$ interaction turns. This sliding-window memory formulation provides a minimal yet realistic abstraction of long-horizon deployment under practical computational and memory constraints.

**Output structure.** The current AI systems (Yang et al., 2025) often remove intermediate reasoning traces from the agent's history to reduce context length and mitigate exposure bias. However, we argue that preserving the *local flow of reasoning* is crucial for coherent long-horizon behavior, particularly when the agent must track subgoals, intermediate decisions, and failure recovery. In this work, we use a structured output format at each step: `<think>...</think>` `REASON:{reason}, ACTION:{action}`, where the `<think>` block contains the CoT reasoning and is not stored in memory, while `reason` is a summary of the reasoning process. The memory $m_t$ stores only the observations, summarized reasoning, and actions from the most recent $K$ steps. This design preserves essential decision context while maintaining computational efficiency and controlling error accumulation.

### C.2. SFT

**SFT prevents reward hacking.** In our initial experiments, we attempted to train agents with RL from scratch, without SFT initialization. This led to severe reward hacking: rather than learning to solve tasks, agents converged to degenerate strategies that avoided penalties through syntactically valid but semantically meaningless actions. For instance, in Sudoku with candidate action features, the agent learned to repeatedly generate candidate sets instead of filling cells with correct values. Introducing SFT initialization successfully resolved this issue by providing valid behavioral priors, demonstrating its essential role for RL.

**SFT initialization with exploration capacity.** While SFT provides essential behavioral priors for RL, aggressive fine-tuning can degrade exploration capabilities (Wu et al., 2025; Chu et al., 2025). SFT models tend to memorize training trajectories, constraining their ability to explore novel behaviors—a critical requirement for effective RL. To mitigate this, we employ a conservative learning rate 5e-6 during SFT (with 4 epochs). Since lower learning rates have been shown to better preserve broader model capabilities (Lin et al., 2025), this approach allows the SFT initialization to provide stable behavioral priors while retaining sufficient exploration capacity for RL training.

## C.3. RL

**Retokenization problem.** Standard LLM inference APIs (i.e., OpenAI Compatible API) are designed for text generation and return strings rather than the token IDs required for RL. This abstraction forces a retokenization step during training, which introduces a dangerous discrepancy because the tokens used to compute log probabilities may not match those actually sampled by the policy (Luo et al., 2025; Research, 2025). Ambiguous tokenization boundaries and tool-call parsing or reformatting can introduce subtle shifts in the token sequence. We find that these shifts act as adversarial noise during reinforcement learning and lead to severe optimization instability. Our experiments confirm that maintaining a strict Tokens-In/Tokens-Out workflow, in which the training pipeline consumes the exact token IDs produced by the inference engine, is critical for preventing policy collapse in multi-turn settings.

**Training-Inference mismatch.** As discussed in Section 2.2.2, many reinforcement learning libraries for LLMs exhibit a training–inference mismatch. In practice, inference pipelines are optimized for high throughput and low latency, often relying on systems such as vLLM (Kwon et al., 2023) or SGLang (Zheng et al., 2024). In contrast, training pipelines prioritize precise and distributed optimization, typically using frameworks such as FSDP (Zhao et al., 2023) or Megatron-LM (Shoeybi et al., 2019). These differences result in distinct execution environments for inference and training. This mismatch naturally introduces off-policy reinforcement learning, since the policy used to generate rollouts differs from the policy being optimized during training. To account for this discrepancy, importance sampling ratios are often required to correct for off-policy updates. However, as we discuss in this work, even small inconsistencies between inference and training can significantly destabilize learning, motivating the need for tighter alignment between the two pipelines.

Recent works (Yao et al., 2025; Liu et al., 2025a) have proposed principled solutions to address the training–inference mismatch and the resulting off-policy instability in RL for LLMs. These approaches analyze importance sampling through the lens of bias–variance trade-offs and show that naive sequence-level correction suffers from exponential variance growth with sequence length, while token-level corrections introduce bias that scales with horizon. To overcome this limitation, Liu et al. (2025a) propose sequence-level truncated and masked importance sampling schemes that explicitly control both variance and bias in long-horizon settings. In particular, they introduce geometric-mean normalization and rejection mechanisms to prevent rare, high-weight trajectories from dominating gradient updates.

Concretely, the importance sampling ratios are defined as:

$$\rho_{\text{seq},t} = \prod_{i=1}^{|y_t|} \frac{\pi_\theta(y_{t,i} \mid x_t, y_{t,<i})}{\mu_{\theta_{\text{old}}}(y_{t,i} \mid x_t, y_{t,<i})}, \quad \rho_{\text{geo},t} = (\rho_{\text{seq},t})^{1/|y_t|}.$$

The geometric-mean ratio $\rho_{\text{geo}}$ normalizes importance weights by sequence length, enabling robust filtering of unreliable or out-of-distribution samples whose weights arise from numerical artifacts or severe distributional shift. For samples that pass this masking step, the sequence-level ratio $\rho_{\text{seq}}$ is then used to control gradient variance while preserving off-policy correction. This two-stage mechanism provides a practical way to stabilize long-horizon agent RL by explicitly accounting for sequence length in importance sampling.

**Codebase: `rllm`.** We base our implementation on `rllm` (Tan et al., 2025), an open-source framework that supports multi-turn reinforcement learning for LLM agents. Specifically, we build on `rllm` v0.2, which uses `verl` v0.5.0 (Sheng et al., 2024) as its backend. During our investigation, we identified several practical issues, including the retokenization problem and training–inference mismatch, which required non-trivial modifications to the original codebase in order to support our proposed RL method. To address these challenges, we extended the framework to preserve token-level consistency throughout the RL pipeline and integrated our implementation with a modified version of `verl` that incorporates solutions for training–inference mismatch. These changes were necessary to ensure stable and correct off-policy optimization in multi-turn settings, and they form a critical part of our experimental infrastructure.

**Experimental setting.** All experiments were conducted using $4 \times$ A100 and $4 \times$ A6000 GPUs. The training and evaluation processes were typically completed within 1 to 3 days. Detailed experimental configurations are provided in Table 3.

*Table 3.* **Hyperparameter configurations used in our experiments.**

| Parameter | Sudoku | Rush Hour |
|---|---|---|
| learning rate | 1e-6 | 1e-6 |
| learning rate scheduler | constant | constant |
| KL loss coefficient | 0.0 | 0.0 |
| KL penalty coefficient | 0.0 | 0.0 |
| maximum response length | 2048 / 4096 (for macro action) | 2048 |
| temperature | 0.8 | 0.8 |
| top_p | 1.0 | 1.0 |
| *Rollout Correction* | | |
| rollout_is | sequence | sequence |
| rollout_is_threshold | 3 | 3 |
| rollout_rs | geometric | geometric |
| rollout_rs_threshold | 1.01 | 1.01 |
| rollout_rs_threshold_lower | 0.995 | 0.995 |
| *Advantage* | | |
| $\gamma$ (discount factor) | 0.995 | 0.995 |
| $\alpha$ | 0.2 | 0.2 |
| *Agent & Environment* | | |
| $H_{\max}$ | 50 | 30 / 20 (for macro action) |
| $K$ turn history | 2 | 2 |

### C.4. Robustness to environment, model scale, and optimizer.

To examine whether the effect of horizon length observed in Sudoku and Rush Hour generalizes beyond our default setup, we conduct additional experiments across three axes: environment, model scale, and optimizer.

**Environment.** We evaluate on WebShop, a web-browsing benchmark requiring multi-step interaction. In this setting, the native action space—consisting of `search` and `choose` operations—serves as the macro-action (horizon-reduced) condition. For the atomic-action condition, we decompose each step into two sequential decisions: first selecting which action type to invoke (`search` or `choose`), and then selecting the corresponding argument. This decomposition doubles the effective horizon length. Interestingly, the atomic-action setting achieves higher initial performance, which we attribute to the additional deliberation step encouraging more careful action selection. However, as training progresses, the horizon-reduced setting improves consistently and ultimately reaches a higher final success rate, in line with our main findings.

**Model scale.** We repeat the Sudoku (L3–L4) experiments using a 4B model to investigate whether increased model capacity alleviates the horizon bottleneck. The results mirror those of the 1.7B model: training collapse occurs under the default horizon, while horizon reduction yields stable and improved performance.

**Optimizer.** To confirm that the observed instability is not specific to our REINFORCE-based optimizer, we repeat the experiments using a GRPO-style method with group-normalized advantages—as opposed to the batch normalization used in our main setup. The results are consistent with our primary findings, suggesting that the horizon bottleneck is optimizer-agnostic.

## D. Additional Analysis

### D.1. Token-Level Gradient Dynamics in RL

Here, we provide a detailed derivation of the token-level gradient dynamics analyzed by Gao et al. (2025), elucidating why negative advantage signals introduce greater optimization instability than their positive counterparts.

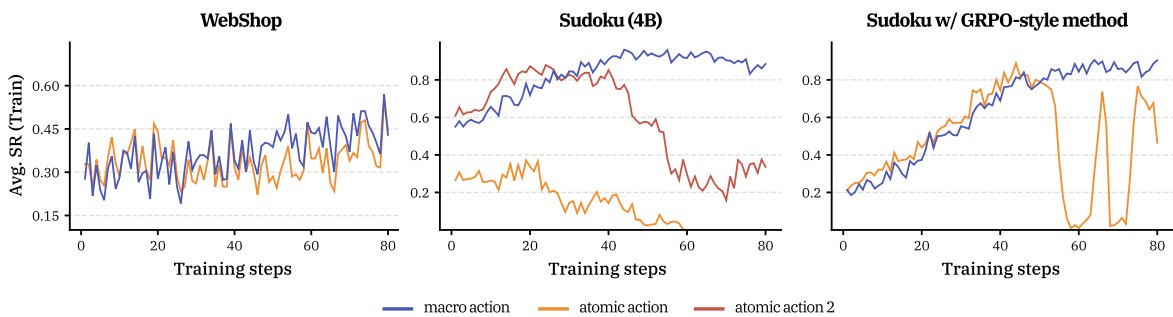

*Figure 10.* **Training dynamics under additional experimental settings.** In all panels, *horizon reduction* refers to training with macro action, while *default* refers to training with atomic action. *Atomic action2* denotes a variant in which the policy is trained with macro action but the environment permits only atomic action, resulting in a longer effective horizon despite the macro-action policy. This setting follows the same protocol as Figure 4.

**Derivation.** Let $z = [z_1, z_2, \ldots, z_{|\mathcal{V}|}]$ denote the logits over a vocabulary $\mathcal{V}$. The policy $\pi_\theta$ computes the probability of a token $(v)$ given context $( (x, y_{i,<t}) )$ via the softmax function:

$$\pi_\theta(v \mid q, y_{i,<t}) = \frac{\exp(z_v)}{\sum_{v' \in \mathcal{V}} \exp(z_{v'})}. \tag{1}$$

To find the sensitivity of the REINFORCE objective $\mathcal{J}$ with respect to any logit ($z_v$), we apply the chain rule:

$$\frac{\partial \mathcal{J}}{\partial z_v} = A_{i,t} \cdot \frac{\partial \log \pi_\theta(y_{i,t})}{\partial \pi_\theta(y_{i,t})} \cdot \frac{\partial \pi_\theta(y_{i,t})}{\partial z_v} = \frac{A_{i,t}}{\pi_\theta(y_{i,t})} \cdot \frac{\partial \pi_\theta(y_{i,t})}{\partial z_v}. \tag{2}$$

Recall the derivative of the softmax function with respect to its logits: $\frac{\partial \pi_\theta(y)}{\partial z_v} = \pi_\theta(y)(\mathbb{I}[v = y] - \pi_\theta(v))$, where $\mathbb{I}$ is the indicator function. Substituting this back into the gradient equation yields:

$$\frac{\partial \mathcal{J}}{\partial z_v} = \frac{A_{i,t}}{\pi_\theta(y_{i,t})} \cdot \pi_\theta(y_{i,t}) \left(\mathbb{I}[v = y_{i,t}] - \pi_\theta(v)\right) \tag{3}$$

$$= A_{i,t} \left(\mathbb{I}[v = y_{i,t}] - \pi_\theta(v)\right). \tag{4}$$

This can be expanded into two cases:

$$\nabla_{z_v} \mathcal{J} = \begin{cases} (1 - \pi_\theta(y_{i,t})) A_{i,t}, & v = y_{i,t}, \\ -\pi_\theta(v) A_{i,t}, & v \neq y_{i,t}. \end{cases} \tag{5}$$

**Analysis of Asymmetry.** Equation 5 reveals a fundamental asymmetry in how the policy is updated based on the sign of the advantage $A_{i,t}$:

- **Positive Advantage:** The gradient increases the logit of the sampled token $y_{i,t}$ and decreases the logits of all unsampled tokens $v \neq y_{i,t}$. This provides a **focused training signal** that reinforces the specific action taken.

- **Negative Advantage:** The gradient decreases the logit of the sampled token but, crucially, **increases the logits of all unsampled tokens** to maintain normalization.

This asymmetry poses a significant challenge in LLMs. The action space corresponds to a massive vocabulary ($|V| \approx 10^5$), yet the set of semantically correct or desirable tokens for any given context is extremely sparse. Consequently, a negative update does not point the model toward the "correct" token; instead, it provides a **diffuse signal** that indiscriminately boosts the probability of tens of thousands of irrelevant tokens. This phenomenon amplifies gradient variance and injects noise into the optimization process, explaining the instability often observed when training with negative constraints or penalties.

*Table 4.* **Examples of sudoku knowledge evaluation sets.**

| Task | Type | N | Example |
|---|---|---|---|
| Rule Knowledge | Multiple-choice | 5 | For 9×9 Sudoku (classic), which rule applies to each 3×3 box?
A. Digits 1–9 must appear without repetition
B. Boxes have no additional rule beyond rows and columns
C. Repetition is allowed in a box if rows are valid
D. A box must contain the digits 1–9 in order
**Answer: A** |
| Technique Definition Knowledge | Multiple-choice | 15 | Choose the correct sudoku solving technique for the following description:
"Three digits that each appear only in the same three cells of a house restrict those cells to those digits."
A) Hidden Triple    B) Hidden Pair    C) Jellyfish    D) Naked Triple
**Answer: A** |
| Technique Identification | Multiple-choice | 10 | What solving technique applies to the following Sudoku situation?
"In row 8, r8c3 and r8c4 are {3,9}, so 3 and 9 are eliminated from other row-8 cells."
A) Swordfish    B) XY-Wing    C) X-Wing    D) Naked Pair
**Answer: D** |

*Table 5.* **Sudoku knowledge evaluation results.**

| Task | **Qwen3-1.7B** (Acc %) | **Qwen3-8B** (Acc %) |
|---|---|---|
| Rule Knowledge | 100.00 | 100.00 |
| Technique Definition Knowledge | 66.67 | 93.33 |
| Technique Identification | 80.00 | 90.00 |

## D.2. Evaluating Sudoku Knowledge

Solving Sudoku puzzles requires domain-specific knowledge, including basic rules and solving techniques. To verify whether our base models possess this knowledge, we conduct targeted knowledge evaluations on Qwen3-1.7B and Qwen3-8B.

We manually construct evaluation sets covering rules and techniques, as detailed in Table 4. As shown in Table 5, both models achieve perfect scores on rule knowledge and demonstrate reasonable familiarity with solving techniques. This level of prior knowledge—accurate rule understanding with moderate technique awareness—is sufficient for our study, as we require models to possess basic domain knowledge as a prerequisite, not to achieve perfect expertise.

## D.3. RL Design Choice

To better understand the design choices underlying stable reinforcement learning, we conduct leave-one-out (LOO) ablation experiments inspired by Khatri et al. (2025). Starting from our full method (**Ours**, top row in Table 6), we systematically ablate both importance sampling functions and advantage formulations.

**Importance sampling ablation.**    We first examine the effect of different IS functions while keeping the advantage definition fixed. Using either sequence-level truncated importance sampling (Seq-TIS) or geometric-mean masked importance sampling (Geo-MIS) alone leads to noticeable performance degradation compared to their combination. Seq-TIS preserves strong pass@4 performance but suffers in average performance, while Geo-MIS exhibits larger drops in both metrics. This suggests complementary roles: Geo-MIS effectively filters unreliable trajectories, while Seq-TIS controls gradient variance during optimization. Combining both mechanisms yields more robust and consistent learning.

**Advantage design ablation.**    We further ablate the advantage formulation while fixing the IS function. Removing normalization or relying solely on raw rewards leads to severe performance degradation, confirming that proper normalization is critical for stable learning. Varying the mixing coefficient $\alpha$ reveals that incorporating step-level rewards improves performance, with moderate values of $\alpha$ providing the best trade-off between trajectory-level supervision and dense feedback.

*Table 6.* **Ablation study of importance sampling functions and advantage design.** We perform leave-one-out ablations starting from our full method (**Ours**), which combines Seq-TIS, Geo-MIS, and a mixed trajectory- and step-level advantage with batch normalization. In the ablation rows, default denotes the corresponding component configuration used in **Ours**; specifically, it refers to using both Seq-TIS and Geo-MIS for IS ablations, and to the mixed advantage $A = \hat{r}^{\text{traj}} + \alpha \hat{r}^{\text{step}}$ with $\alpha = 0.2$ and batch normalization for advantage ablations. Removing or modifying any single component leads to degraded performance, highlighting the importance of jointly designing importance sampling and advantage normalization for stable RL.

| IS Function ($w$) | Advantage | avg@4 | pass@4 |
|---|---|---|---|
| *Ours* | | | |
| Seq-TIS + Geo-MIS | $A = \hat{r}^{\text{traj}} + \alpha \hat{r}^{\text{step}}$ ($\alpha = 0.2$, Batch Normalization) | **96.0** | **97.6** |
| *Ablation of IS Function* | | | |
| Seq-TIS | default | 91.4 | **97.6** |
| Geo-MIS | default | 89.4 | 96.0 |
| *Ablation of Advantage* | | | |
| default | $A = \hat{r}^{\text{traj}} + \alpha \hat{r}^{\text{step}}$ ($\alpha = 0.2$, Group Normalization for $r^{\text{traj}}$) | 83.4 | 95.2 |
| default | $A = r^{\text{traj}} + \alpha r^{\text{step}}$ ($\alpha = 0.2$, No normalization) | 44.4 | 77.6 |
| default | $A = \hat{r}^{\text{traj}} + \alpha \hat{r}^{\text{step}}$ ($\alpha = 0.0$) | 91.4 | 96.0 |
| default | $A = \hat{r}^{\text{traj}} + \alpha \hat{r}^{\text{step}}$ ($\alpha = 0.5$) | 93.0 | **97.6** |
| default | $A = \hat{r}^{\text{traj}} + \alpha \hat{r}^{\text{step}}$ ($\alpha = 1.0$) | 95.8 | **97.6** |

Overall, these results demonstrate that stable long-horizon RL requires careful co-design of importance sampling and advantage normalization rather than relying on either component in isolation.

### D.4. Horizon Generalization

**What is step accuracy?** Step accuracy measures the probability of successfully performing a single step, independent of multi-step execution ability (Sinha et al., 2025). This metric is valuable for analyzing failure modes in long-horizon tasks, specifically, it helps distinguish whether failures stem from errors at the step level or from error propagation and horizon-related challenges. However, defining and computing step accuracy in practice is non-trivial, as not all actions can be clearly classified as correct or incorrect. For example, it is unclear whether repeating meaningless actions or taking suboptimal but valid actions should be considered successful steps.

**Step accuracy in Sudoku.** Fortunately, our Sudoku environment simplifies this problem. In our setting, the only available action is assigning a value to a specific cell, and each cell has a unique correct value. We therefore compute step accuracy as shown in Figure 8 (right): steps that assign the correct value are classified as successes, while all others are classified as failures.

**Limitations in Rush Hour.** In contrast, computing step accuracy in Rush Hour is challenging. It is difficult to determine whether each vehicle movement contributes meaningfully to the solution or represents an unnecessary detour. Consequently, unlike Sudoku, we report only the average success rate as a function of goal distance for Rush Hour in Figure 11.

**Horizon generalization without technique generalization.** Sudoku solving strategies can be formalized as techniques—recognizable cell configuration patterns such as Naked Singles that enable deductive reasoning steps. While our previous results demonstrate that RL-trained agents exhibit strong horizon generalization (Figure 8), we find that this generalization does not extend to the level of solving techniques.

To analyze this limitation, we first categorize Sudoku techniques according to their reasoning complexity. We define three levels of technique difficulty:

- *Easy:* Techniques that directly determine a cell's value from a single remaining candidate. *Full House, Naked Single*.

- *Medium:* Techniques that require local reasoning over multiple candidates or simple pattern-based eliminations. *Hidden Single, Locked Candidates Type 1 (Pointing), Locked Candidates Type 2 (Claiming), Naked Pair, Naked Triple, Hidden Pair, Hidden Triple, Locked Pair, Locked Triple, X-Wing, Uniqueness Test 1, Uniqueness Test 2, Uniqueness Test 3, Uniqueness Test 4, Uniqueness Test 6*.

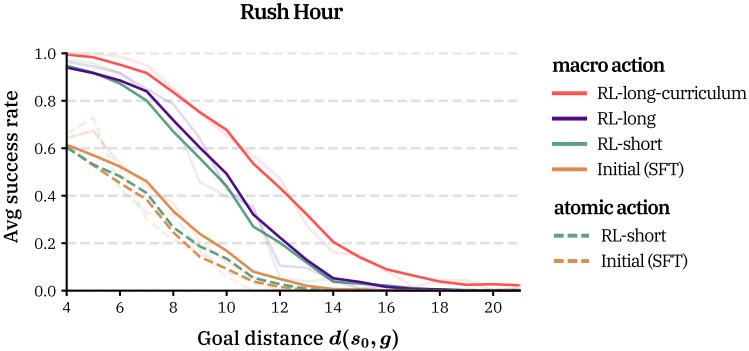

*Figure 11.* **Successes rates and goal distance for Rush Hour.** *RL-short* trains on $4 \leq d(s_0, g) \leq 9$, *RL-long* trains on $10 \leq d(s_0, g) \leq 12$, and *RL-long-curriculum* first trains on short horizons then continues on long horizons.

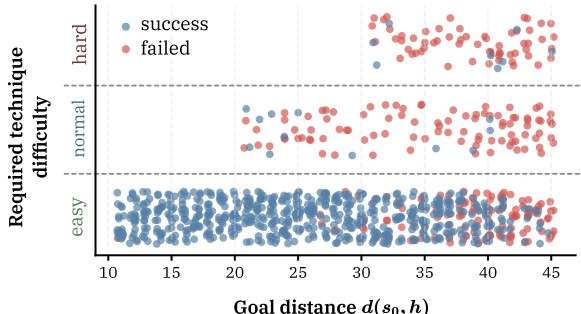

*Figure 12.* **Evaluation of horizon and technique generalization in Sudoku.** Generalization holds within seen (*easy*) techniques but breaks under increased technique difficulty (*medium* and *hard*). Points are jittered for visualization.

- ***Hard:*** Techniques that involve long-range dependency tracking, chained reasoning, or complex candidate propagation across multiple units. *Naked Quadruple, Hidden Rectangle, Avoidable Rectangle Type 1, Swordfish, Simple Colors Trap, Skyscraper, 2-String Kite, Empty Rectangle, XY-Wing, XYZ-Wing, W-Wing, Remote Pair, AIC, Grouped AIC, Continuous Nice Loop, Discontinuous Nice Loop, Grouped Continuous Nice Loop, Grouped Discontinuous Nice Loop, XY-Chain, Almost Locked Set Chain, Almost Locked Set XY-Wing, Almost Locked Set XZ-Rule, Sue de Coq, Turbot Fish, Finned X-Wing, Sashimi X-Wing, Finned Swordfish, Sashimi Swordfish, Finned Jellyfish, Finned Franken Swordfish, Multi Colors 1, Brute Force.*

A limitation of our original Sudoku evaluation benchmark (L1-L7) is that it consists almost exclusively of puzzles solvable using *easy* techniques. As a result, evaluating generalization across technique difficulty is not possible with this benchmark alone. To address this, we expand the evaluation set by additionally sampling puzzles that are solvable by a stronger reference model, GPT-5-mini with pass@4. This procedure introduces puzzles that require more advanced reasoning patterns while remaining within a solvable regime.

Figure 12 reports the performance of our RL model trained on L3-L4 (21-30 goal distance) puzzles when evaluated on this expanded benchmark. The results reveal a clear distinction between horizon and technique generalization. When puzzles require techniques comparable to those seen during training, the RL agent generalizes effectively to longer horizons (*easy*). However, performance degrades sharply as the required techniques become more difficult, even when the horizon length remains within the training distribution (*medium* and *hard*).

This observation suggests that while RL can learn to extend horizon within a familiar deductive framework, it struggles to acquire fundamentally new reasoning primitives. In this sense, our findings align with prior works (Yuan et al., 2025; Yue et al., 2025) showing that RL fine-tuning typically amplifies and recombines capabilities already present in the base model, rather than enabling qualitatively novel forms of reasoning beyond its pre-trained repertoire.

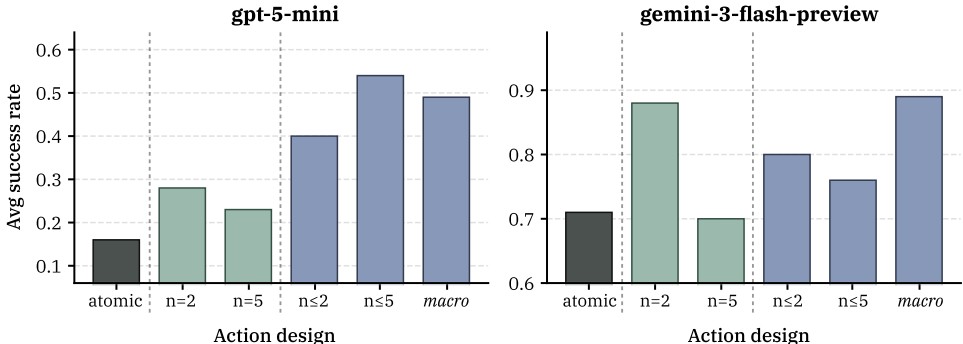

*Figure 13.* **Effect of macro action design on frontier models.** Average success rates for GPT-5-mini and Gemini-3-Flash-Preview under different action designs, including atomic actions, fixed-length macro actions ($n$=2, 5), and flexible macro actions ($n \leq k$ or unbounded). Flexible macro actions are generally beneficial across models, although their performance ceiling varies by model. Notably, for Gemini-3-Flash-Preview, a small fixed macro length ($n = 2$) performs competitively, while larger fixed lengths ($n = 5$) degrade performance, suggesting that excessive action aggregation can be detrimental and that modest horizon reduction may suffice depending on model capacity.

*Table 7.* **Sudoku evaluation across training and test horizons.** We report pass@4 and average success rate (avg@4) for models trained under different settings (macro vs. atomic) and training horizons $d_{\text{train}}(s_0, g)$, evaluated across increasing test goal distances $d_{\text{test}}(s_0, g)$. Results show that macro action RL and subgoal decomposition substantially improve performance and stability on long-horizon tasks, while atomic-action RL degrades or collapses as the horizon increases.

| action | train | $d_{\text{train}}(s_0, g)$ | | $d_{\text{test}}(s_0, g)$ | | | | | | |
|---|---|---|---|---|---|---|---|---|---|---|
| | | | | L1 11–15 | L2 16–20 | L3 21–25 | L4 26–30 | L5 31–35 | L6 36–40 | L7 41–45 |
| **macro** | Base | - | pass@4 (%) | 98.00 | 90.00 | 67.00 | 28.00 | 8.00 | 0.00 | 0.00 |
| | | | avg@4 (%) | 66.50 | 44.00 | 23.75 | 6.25 | 0.50 | 0.25 | 0.00 |
| | Initial SFT | 11-20 | pass@4 (%) | 98.00 | 90.00 | 67.00 | 28.00 | 8.00 | 0.00 | 0.00 |
| | | | avg@4 (%) | 69.50 | 48.75 | 26.25 | 8.75 | 2.00 | 0.00 | 0.00 |
| | RL | 11-20 | pass@4 (%) | **100** | **100** | 97.00 | 84.00 | 61.00 | 31.00 | 10.00 |
| | | | avg@4 (%) | 95.75 | 86.50 | **68.75** | 41.75 | 26.25 | 8.25 | 2.50 |
| | | 21-30 | pass@4 (%) | 99.00 | **100.00** | **98.00** | **91.00** | **85.00** | **61.00** | **38.00** |
| | | | avg@4 (%) | **97.00** | **95.50** | 87.25 | **69.50** | **54.00** | **30.75** | **14.00** |
| **atomic** | Initial SFT | 11-20 | pass@4 (%) | 90.00 | 64.00 | 26.00 | 7.00 | 2.00 | 0.00 | 0.00 |
| | | | avg@4 (%) | 58.50 | 27.75 | 7.75 | 2.00 | 0.50 | 0.00 | 0.00 |
| | RL | 11-20 | pass@4 (%) | **100** | 91.00 | 75.00 | 39.00 | 14.00 | 2.00 | 0.00 |
| | | | avg@4 (%) | 86.50 | 65.25 | 43.25 | 13.75 | 4.00 | 0.50 | 0.00 |
| | | 21-30 (before collpase) | pass@4 (%) | 98.00 | 79.00 | 58.00 | 27.00 | 11.00 | 0.00 | 0.00 |
| | | | avg@4 (%) | 81.50 | 54.00 | 29.75 | 8.75 | 3.25 | 0.00 | 0.00 |
| | RL (subgoal decomposition) | 21-30 | pass@4 (%) | **100** | 99.00 | 93.00 | 69.00 | 31.00 | 12.00 | 0.00 |
| | | | avg@4 (%) | 84.50 | 73.50 | 51.00 | 28.75 | 11.50 | 3.00 | 0.00 |

*Table 8.* **Rush Hour evaluation across training and test horizons.** Results show that training with macro action, particularly when combined with horizon-aware curriculum training, substantially improves performance and generalization on longer horizons.

| action | train | $d_{\text{train}}(s_0, g)$ | | $d_{\text{test}}(s_0, g)$ | | | | | |
|--------|-------|------|------|------|------|------|------|------|------|
| | | | | **4–6** | **7–9** | **10–12** | **13–15** | **16–18** | **19–21** |
| **macro** | Base | – | pass@4 (%) | 38.37 | 5.68 | 0.00 | 0.00 | 0.00 | 0.00 |
| | | | avg@4 (%) | 11.82 | 1.52 | 0.00 | 0.00 | 0.00 | 0.00 |
| | SFT | 4–12 | pass@4 (%) | 94.19 | 71.59 | 19.78 | 3.00 | 0.00 | 0.00 |
| | | | avg@4 (%) | 60.51 | 34.28 | 7.01 | 1.00 | 0.00 | 0.00 |
| | RL | 4–9 | pass@4 (%) | **100** | 87.5 | 58.24 | 12.00 | 3.00 | 0.00 |
| | | | avg@4 (%) | 94.53 | 67.90 | 30.36 | 4.25 | 0.75 | 0.00 |
| | | 10–12 | pass@4 (%) | **100** | 92.05 | 64.84 | 18.00 | 3.00 | 1.00 |
| | | | avg@4 (%) | 93.65 | 76.56 | 32.14 | 5.50 | 0.75 | 0.25 |
| | RL (curriculum) | 10–12 | pass@4 (%) | **100** | **97.73** | **84.62** | **43.00** | **15.00** | **8.00** |
| | | | avg@4 (%) | **99.42** | **85.65** | **56.18** | **20.25** | **5.00** | **2.25** |
| **atomic** | Base | – | pass@4 (%) | 24.42 | 7.95 | 0.00 | 0.00 | 0.00 | 0.00 |
| | | | avg@4 (%) | 6.30 | 1.23 | 0.00 | 0.00 | 0.00 | 0.00 |
| | SFT | 4–12 | pass@4 (%) | 89.53 | 53.41 | 15.38 | 0.00 | 0.00 | 0.00 |
| | | | avg@4 (%) | 58.62 | 27.70 | 4.95 | 0.00 | 0.00 | 0.00 |
| | RL | 4–9 | pass@4 (%) | 93.02 | 45.45 | 6.59 | 1.00 | 0.00 | 0.00 |
| | | | avg@4 (%) | 60.56 | 21.45 | 2.34 | 0.25 | 0.00 | 0.00 |

**Prompt (Sudoku)**

You are a professional sudoku solver. You are given a sudoku board and you need to solve it. You can solve the puzzle over multiple turns, so it's not necessary to outline the full solution at once.

## Format Explanation
Coordinates:
- We will use rXcY coordinates. For example, r1c1 is the top-left cell at row 1 column 1, r1c2 is the cell to the right at row 1 column 2, r2c1 is the cell below at row 2 column 1, and so on.

Representation of the board:
- Initial board values (given numbers) are represented as value (e.g., 4, 7).
- Empty cells are represented as '.' (e.g., '.') and your value will be placed in the cell (e.g., '3').

### Output Requirements
1. Thought: Provide a detailed, step-by-step reasoning process explaining your thought process in solving the task.
2. Reason: Give a concise explanation summarizing the key logic behind your action(s).
3. Action: List the concrete solving actions you want to take, each on its own line, wrapped in triple backticks.

### Output Format
You must generate your thought, reason and action in the following format:
<think>\n[Your thought process in solving the task.]\n</think>
REASON: [Your reason for the action(s)]
ACTION: ```\n[one or more actions, each on its own line]\n```

## Environment Information
### Available Actions
1. Value Setting: ```value(digit, rXcY)```
- Assign a confirmed digit (1−9) to a specific cell.
- Must obey all Sudoku constraints (row/column/box/rules).
- Used only when the cell's value is fully deduced.

### Goal
{goal}

### Size
{board_size}

### Rules
{rules}

Here is the recent history:
{history}

## Current Observation
{current_observation}

The maximum number of steps remaining is {remain_turn}.

Enclose your detailed reasoning process within <think> and </think> tags. After that, summarize your reasoning (not exceed 8 sentences) following the 'REASON:' tag. Finally, produce your action after the 'ACTION:' tag, wrapping it in triple backticks (```).

*Figure 14.* **Prompt used for Sudoku experiments.**

**Prompt (Rush Hour)**

You are a professional Rush Hour solver. You are given a Rush Hour board and you need to solve it.

## Format Explanation
Representation of the board:
The board is given as a rectangular grid of characters.
Each non-'x' character represents part of a vehicle.
Identical letters belong to the same vehicle and occupy contiguous cells either horizontally or vertically.
- 'x' denotes a wall (immovable and impassable).
- '.' denotes an empty cell.
- Each uppercase letter (A, B, C, ...) denotes a car.
- The special car AA is the target car that must reach the exit.
- The exit is located on the right boundary of the row containing AA.
- The puzzle is solved when AA occupies the exit cell.

### Output Requirements
1. Thought: Provide a detailed, step-by-step reasoning process explaining your thought process in solving the task.
2. Reason: Give a concise explanation summarizing the key logic behind your action(s).
3. Action: List the concrete solving actions you want to take, each on its own line, wrapped in triple backticks.

### Output Format
You must generate your thought, reason and action in the following format:
<think>\n[Your thought process in solving the task.]\n</think>
REASON: [Your reason for the action(s)]
ACTION: ```\n[one or more actions, each on its own line]\n```

## Environment Information
### Available Actions
1. Move: ```move(vehicle_name, direction, num_moves)```
- Move a vehicle from one cell to another.
- vehicle_name: name of the vehicle to move (e.g. 'A', 'B', 'C', ...)
- direction: direction of movement ('up', 'down', 'left', 'right')
- num_moves: number of cells to move (default: 1)

### Goal
{goal}

### Rules
{rules}

Here is the recent history:
{history}

## Current Observation
{current_observation}

The maximum number of steps remaining is {remain_turn}.

Enclose your detailed reasoning process within <think> and </think> tags. After that, summarize your reasoning (not exceed 8 sentences) following the 'REASON:' tag. Finally, produce your action after the 'ACTION:' tag, wrapping it in triple backticks (```).

*Figure 15.* **Prompt used for Rush Hour experiments.**

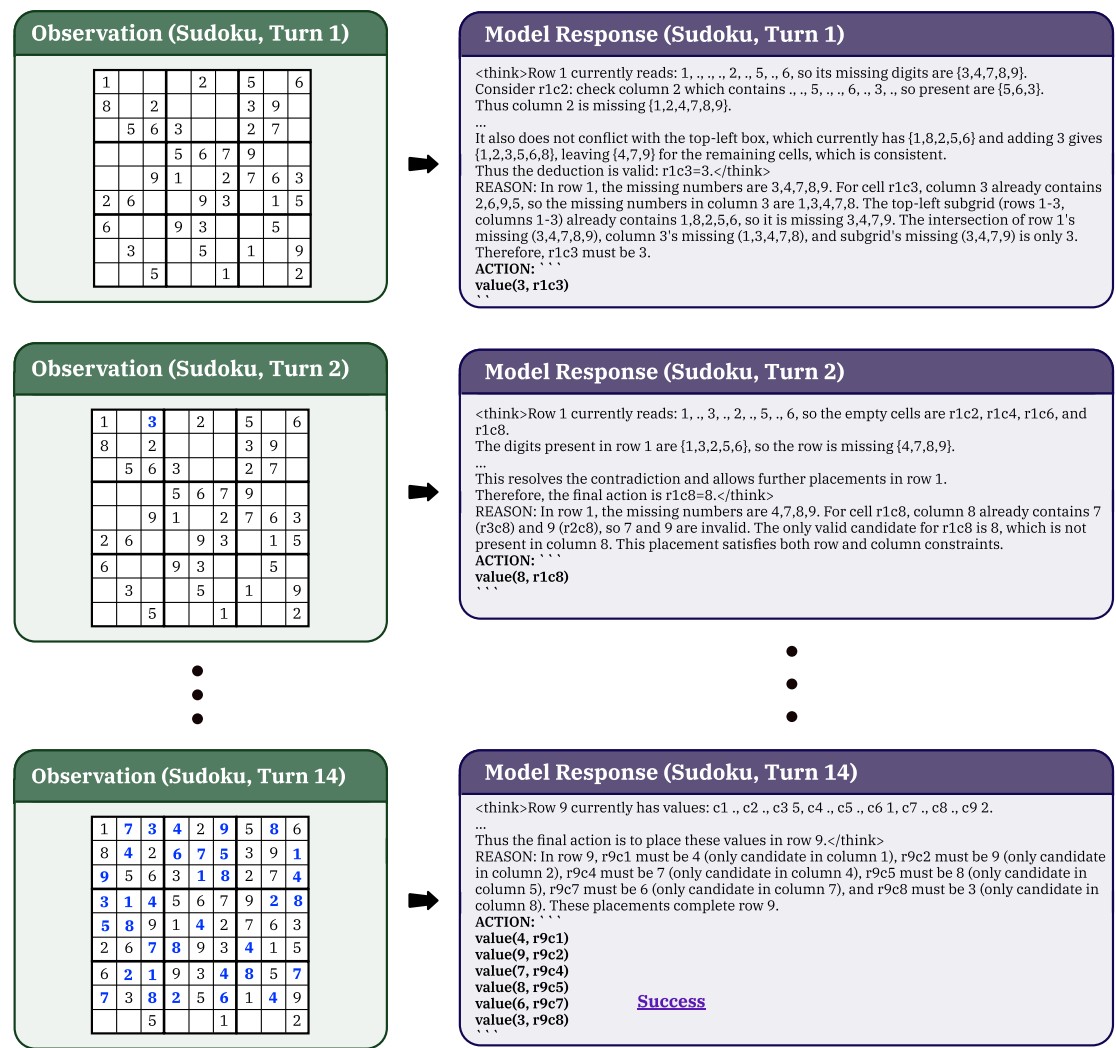

*Figure 16.* **Case study for our RL model in Sudoku (successful case).**

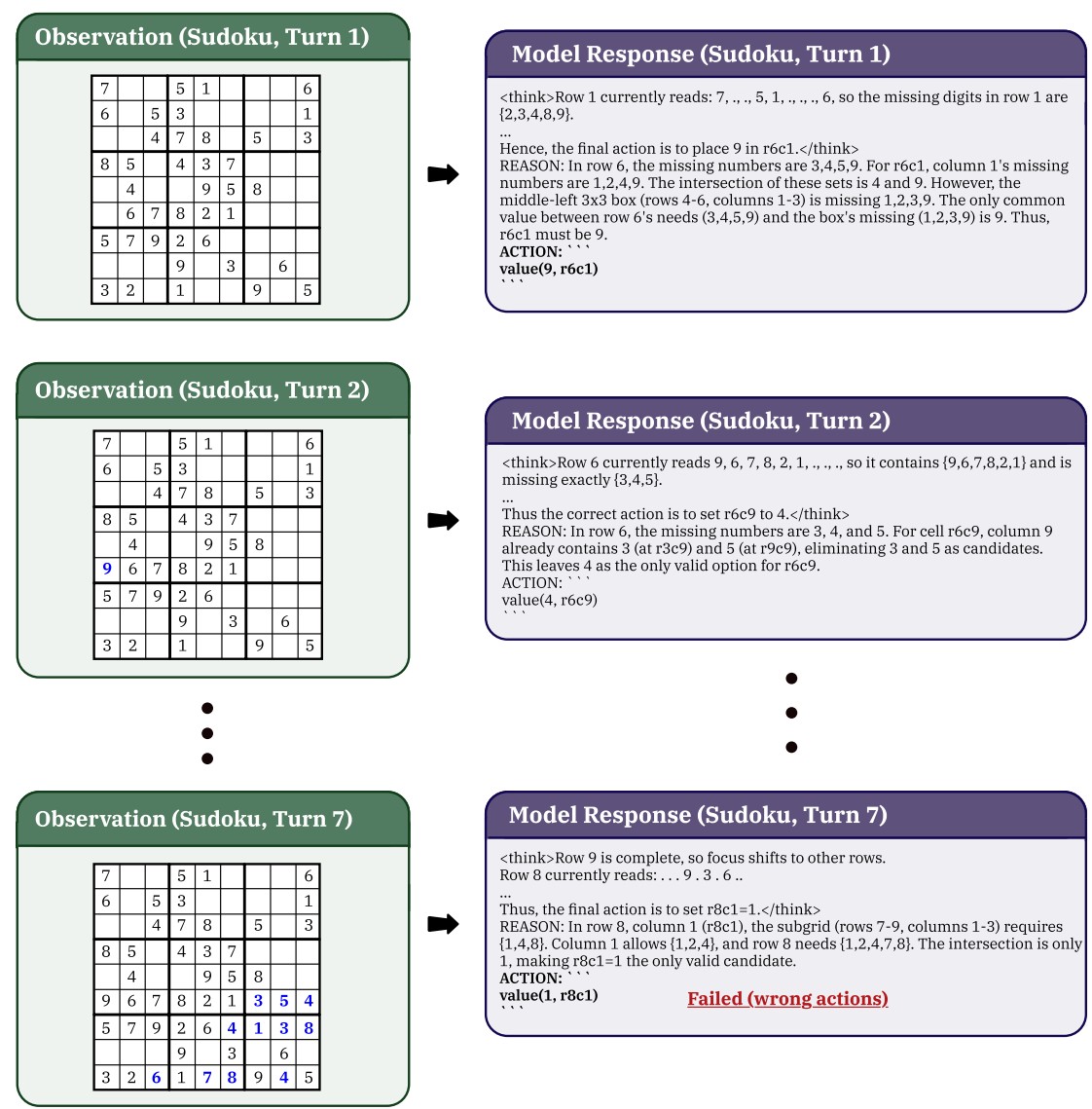

*Figure 17.* **Case study for our RL model in Sudoku (failed case).**

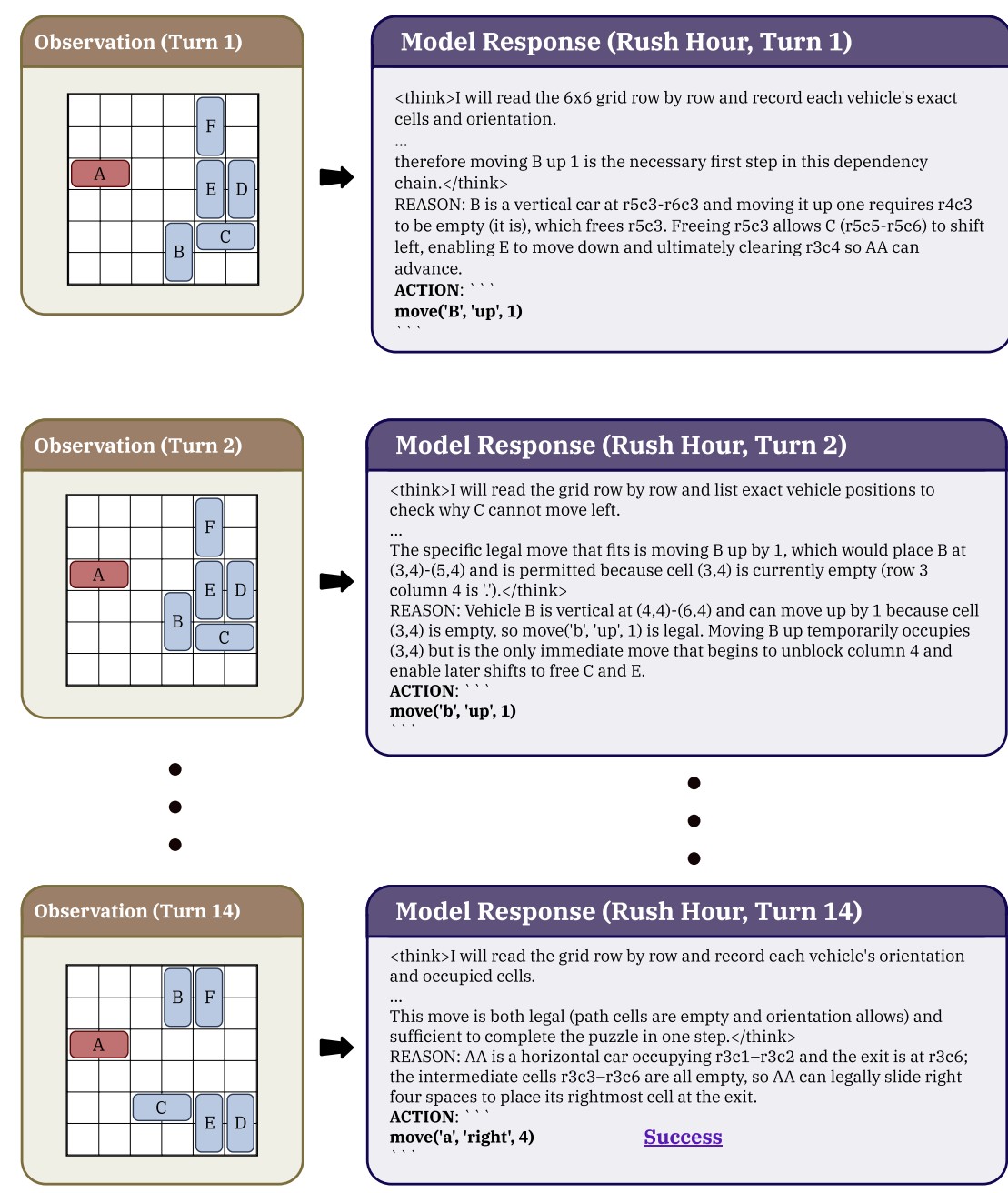

*Figure 18.* **Case study for our RL model in Rush Hour (successful case).**

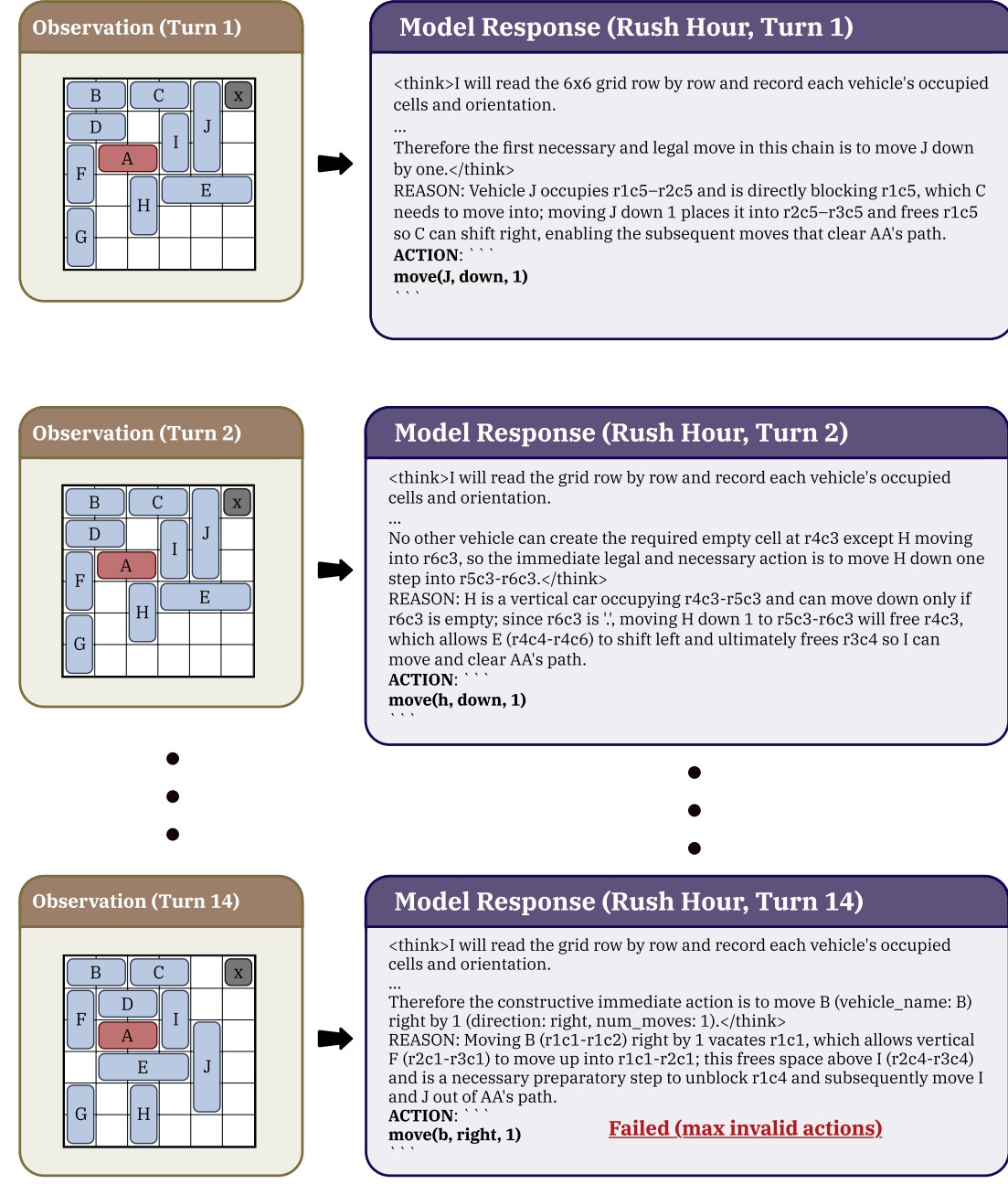

**Observation (Turn 1)**

**Model Response (Rush Hour, Turn 1)**

<think>I will read the 6x6 grid row by row and record each vehicle's occupied cells and orientation.
...
Therefore the first necessary and legal move in this chain is to move J down by one.</think>
REASON: Vehicle J occupies r1c5–r2c5 and is directly blocking r1c5, which C needs to move into; moving J down 1 places it into r2c5–r3c5 and frees r1c5 so C can shift right, enabling the subsequent moves that clear AA's path.
**ACTION**: ```
**move(J, down, 1)**
```

**Observation (Turn 2)**

**Model Response (Rush Hour, Turn 2)**

<think>I will read the grid row by row and record each vehicle's occupied cells and orientation.
...
No other vehicle can create the required empty cell at r4c3 except H moving into r6c3, so the immediate legal and necessary action is to move H down one step into r5c3-r6c3.</think>
REASON: H is a vertical car occupying r4c3–r5c3 and can move down only if r6c3 is empty; since r6c3 is '.', moving H down 1 to r5c3-r6c3 will free r4c3, which allows E (r4c4-r4c6) to shift left and ultimately frees r3c4 so I can move and clear AA's path.
**ACTION**: ```
**move(h, down, 1)**
```

**Observation (Turn 14)**

**Model Response (Rush Hour, Turn 14)**

<think>I will read the grid row by row and record each vehicle's occupied cells and orientation.
...
Therefore the constructive immediate action is to move B (vehicle_name: B) right by 1 (direction: right, num_moves: 1).</think>
REASON: Moving B (r1c1-r1c2) right by 1 vacates r1c1, which allows vertical F (r2c1-r3c1) to move up into r1c1-r2c1; this frees space above I (r2c4-r3c4) and is a necessary preparatory step to unblock r1c4 and subsequently move I and J out of AA's path.
**ACTION**: ```
**Failed (max invalid actions)**
**move(b, right, 1)**
```

*Figure 19.* **Case study for our RL model in Rush Hour (failed case).**

