# OpenReview forum: "On Training Large Language Models for Long-Horizon Tasks: An Empirical Study of Horizon Length"
_ICML.cc/2026/Conference — ICML 2026 regular_

### Official Review · Reviewer_pv3n · 2026-03-12

**Soundness:** 2
**Presentation:** 2
**Significance:** 2
**Originality:** 3
**Overall Recommendation:** 3
**Confidence:** 3

**Summary:**

The paper present a formalism for understanding the role of task solution length (horizon length) in the training dynamics. Authors construct two deterministic environments intended to isolate the length of the horizon from the reasoning complexity. They show training collapse for long horizon tasks in these environments, and propose two methods to mitigate it: macro actions and subgoal decomposition. The authors provide evidence of the positive influence of the proposed approaches on training dynamics and final scores. They also demonstrate the training collapse induced by the horizon length in a controlled environment. The authors also propose a curriculum strategy for training on long horizon tasks, and show that it can be effective in mitigating the training collapse. Finally, the paper shows the generalization across different horizon lengths.

**Compliance With Llm Reviewing Policy:**

Affirmed.

**Final Justification:**

Despite the proposed framework seems novel and provides new insights, it's further development for application to real-world tasks appears to be at least non-trivial. The main concerns raised in the review are addressed, however, both subgoal decomposition and design of macro actions approaches remain questionable for real-world tasks. Therefore, the score is kept 3.

**Key Questions For Authors:**

1. What would be a macro/micro action in SWE-bench like environments?
2. How are the proposed methods different from the standard hierarchical RL approaches?
3. For macro actions, how the gradient is propagated to the probability distributions of individual tokens?

**Limitations:**

yes

**Strengths And Weaknesses:**

Strengths:
- clean controlled experimental design, varying horizon, holding reasoning complexity fixed
- new formalism, that is potentially useful in distinguishing different phenomena the task solution length induce
- demonstration of training collapse for long tasks as reproducible and diagnosable problem

Weaknesses:
- limited scope of the datasets
- key findings are obtained on the synthetic datasets with simple deterministic environments
- subgoal decomposition is not available for the majority of realistic scenarios, e.g. for the environments like SWE agents, there is no intermediate reward
- the curriculum strategy requires a known, measurable difficulty metric (goal distance) — in real tasks there is no obvious oracle for horizon length
- macro actions seems to be already used by default by all modern agents, while, at the same time, they are not applicable to reactive environments e.g. robotics
- the definition of horizon seems to hard to apply to real-world tasks

---

> ### Author Rebuttal · Authors · 2026-03-31
>
> We thank the reviewer for the careful reading and thoughtful feedback. We first address the three questions, as they clarify the main ideas and several concerns about realism and scope.
> ## Questions
> > What would be a macro/micro action in SWE-bench like environments?
>
> In a SWE-bench-like setting, micro actions are low-level operations such as file access, code edits, or test execution. A macro action bundles several such operations into a single decision, e.g., submitting a patch or running a script that performs search, edit, and validation. SWE-agent [1] illustrates this by replacing raw shell commands with higher-level commands. This reduces the number of decisions: a bug fix that requires many micro actions can be completed in a few macro actions, directly reducing the effective horizon.
>
> > How are the proposed methods different from the standard HRL approaches?
>
> Unlike HRL, we do not learn a separate high-level policy. However, we view this as secondary: our contribution lies in identifying horizon length as a fundamental bottleneck and showing it can be mitigated through action space design and divide-and-conquer structuring. This is complementary to hierarchical approaches and provides a controlled characterization of why decomposition improves stability.
>
> > For macro actions, how the gradient is propagated to the probability distributions of individual tokens?
>
> Macro actions are treated as single environment steps. All tokens receive the same step-level reward and advantage, with no additional intra-action credit assignment. The benefit comes from reducing the number of steps and shortening the effective horizon, not from modifying gradient computation.
>
> [1] SWE-agent: Agent-Computer Interfaces Enable Automated Software Engineering
>
> ---
> ## W1. Analysis Under Deterministic Conditions
> Our controlled environments isolate the effect of horizon length without confounds, allowing us to study how the effective horizon influences training stability. We also agree that validating generalization beyond such controlled settings is an important and constructive direction for strengthening our claims. To demonstrate broader applicability, we conduct additional experiments in WebShop [1].
>
> **[Results (Avg@4) on WebShop]**
>
> |  | default | w/ horizon reduction |
> | :---- | :---- | :---- |
> | epoch 1 | **32.8** | 28.4 |
> | epoch 2 | 27.3 | **31.4** |
> | epoch 3 | 32.8 | **33.6** |
> | epoch 4 | 38.5 | **42.0** |
>
> These results are consistent with our core findings, providing initial evidence that horizon reduction generalizes beyond our primary setting. We agree that further experiments in richer environments remain valuable future work.
>
> [1] Webshop: Towards scalable real-world web interaction with grounded language agents.
>
> ---
> ## W2. Limitation in Generalizing to Real-World Task
> **Macro Action.**
> Several recent academic works for training LLM agents consider agent settings where the interaction interface remains at the level of atomic actions, such as direct API calls or primitive GUI operations [1–2]. Our point is complementary: action space design itself is an important and under-discussed lever for stabilizing long-horizon learning.
> For web agents, low-level interactions can often be replaced by higher-level API calls [3]. In robotics, action chunking reduces compounding errors by predicting short action sequences [4–5]. Although our experiments are conducted in controlled settings, we argue that the principle of horizon reduction extends to these domains.
>
> **Subgoal Decomposition.**
> Our claim is not that real-world tasks always provide oracle step-level rewards, but that many long-horizon tasks can be restructured into shorter, verifiable segments. The global goal can often be decomposed into subgoals so that each provides a local learning signal, instantiated via planner-generated milestones, checklists, or process-reward formulations.
>
> **Curriculum.**
> An exact oracle for horizon length is rarely available, but our results suggest precision is unnecessary. In Rush Hour, a coarse two-stage curriculum already yields clear gains, indicating that only a rough monotonic proxy for difficulty is needed such as solved trajectory length, number of tool calls, or simple binning. This is consistent with concurrent work [6–7].
>
> [1] Reinforcement Learning for Long-Horizon Interactive LLM Agents
> [2] Agent-R1: Training Powerful LLM Agents with End-to-End Reinforcement Learning
> [3] Beyond Browsing: API-Based Web Agents
> [4] Learning Fine-Grained Bimanual Manipulation with Low-Cost Hardware
> [5] RoboAgent: Generalization and Efficiency in Robot Manipulation via Semantic Augmentations and Action Chunking
> [6] Thinking vs. Doing: Agents that Reason by Scaling Test-Time Interaction
> [7] AgentGym-RL: Training LLM Agents for Long-Horizon Decision Making through Multi-Turn Reinforcement Learning

---

> > ### Author Rebuttal · Reviewer_pv3n · 2026-04-04
> >
> > Thank authors for a detailed response. I believe that the most of the questions are adequately addressed in the rebuttal, however, I still find the evidence supporting the scalability of the presented framework to the real-world tasks limited because of non-trivial nature of the token-wise generation of the language models.

---

> > > ### Author Response · Authors · 2026-04-07
> > >
> > > We are glad that most of our earlier responses were helpful.
> > >
> > > We would like to clarify one point that we believe addresses your remaining concern. In our ablation (Fig. 4), we keep the macro-action policy representation fixed but restore a long interaction horizon by forcing execution at the level of single atomic actions. Training becomes unstable again under this condition. This demonstrates that the benefit of macro actions stems not merely from a richer output representation, but specifically from reducing delayed credit assignment across environment-level steps. From this perspective, realistic LLM agents face two coupled challenges: token-level generation and interaction-level horizon. Our paper provides controlled evidence that the latter is an independent bottleneck-one that persists even when the former is held fixed.
> > >
> > > Beyond the primary setting, we provide additional evidence spanning WebShop, a POMDP environment, a GRPO-style optimizer, and a 4B model, all of which are qualitatively consistent with the main findings. We acknowledge that validation on real-world agents (e.g., coding agent) remains an important open direction, and we will make this limitation explicit in the camera-ready version. We respectfully submit that the paper offers a meaningful contribution: **a precise characterization of effective horizon as a training bottleneck, paired with simple mitigation principles that generalize beyond the primary controlled setting.**

---

### Official Review · Reviewer_PBJi · 2026-03-13

**Soundness:** 3
**Presentation:** 3
**Significance:** 3
**Originality:** 2
**Overall Recommendation:** 4
**Confidence:** 3

**Summary:**

This paper argues that horizon length is a key challenge when training LLM agents with RL, regardless of how complex the task is. They run controlled experiments on Sudoku and Rush Hour tasks, changing the horizon but keeping the reasoning difficulty the same. They show that longer horizons by themselves lead to unstable training and even collapse. To address this, they propose 'horizon reduction' using macro actions and decomposing tasks into subgoals as a design principle to stabilize training. The results also show that models trained on shorter horizons still perform well on longer ones during testing.

**Compliance With Llm Reviewing Policy:**

Affirmed.

**Key Questions For Authors:**

See weaknesses.

**Limitations:**

Yes.

**Strengths And Weaknesses:**

Strengths:

1. The experimental design is rigorous. Separating task complexity from horizon length is a well-known challenge in typical agent benchmarks. They create a controlled setting by using procedurally generated tasks, such as Sudoku and Rush Hour. The baseline comparisons between atomic actions, fixed-length macro actions, and flexible macro actions are clear and well-supported by the empirical data.

2. Figure 1 gives a clear overview of the paper’s structure: problem formulation of isolating horizon from complexity -> theoretical analysis of mapping complexity and credit assignment -> proposed empirical solutions.  The paper’s logic is straightforward and easy to follow.

3. This work addresses a bottleneck in current agentic RL research. By proving that horizon length is an independent axis of failure, the paper provides a strong diagnostic tool for practitioners. Additionally, the findings on horizon generalization and curriculum learning provide insights for researchers training agentic RL.


Weaknesses:

1. The entire empirical story is based on two text-based puzzle games with a 1.7B model. While the authors justify this choice, it limits the paper's ability to support their broad claims made in the discussion and conclusion. Current long-horizon LLM tasks are usually partially observable and noisy (like coding tasks, web broswing). The Sudoku and Rush Hour benchmarks have discrete and fully observable state spaces with deterministic transitions. The paper would be significantly stronger with at least one experiment on a more realistic agent benchmark.

2. The paper uses a single RL algorithm (REINFORCE with off-policy corrections and batch normalization). It is unclear whether the observed empirical findings hold under other more stable RL methods (such as GRPO, and PPO). The paper would benefit from findings not specific to the optimization setup.

3. There is no comparison with reward shaping or dense reward signals beyond the subgoal decomposition experiment. Process reward models (PRMs) are discussed in the related work but are not compared as baselines / alternative methods in experiments.

---

> ### Author Rebuttal · Authors · 2026-03-31
>
> ## W1. Analysis Under Deterministic Conditions
> We thank the reviewer for this concrete and constructive feedback. We fully agree that broader empirical validation strengthens the paper's claims, and we have conducted additional experiments to address this directly.
>
> Specifically, we evaluate on WebShop \[1\], a widely adopted benchmark in recent RL for LLM agent works \[2-4\], which involves partially observable, noisy, and multi-step decision-making. Unlike Sudoku and Rush Hour, WebShop features natural language observations, stochastic transitions, and open-ended action spaces, making it a substantially more realistic testbed. We additionally report Sudoku results with a 4B model to address concerns about model scale. All results are presented below.
>
> **[Results (Avg@4) on WebShop]**
>
> |  | default | w/ horizon reduction |
> | :---- | :---- | :---- |
> | epoch 1 | **32.8** | 28.4 |
> | epoch 2 | 27.3 | **31.4** |
> | epoch 3 | 32.8 | **33.6** |
> | epoch 4 | 38.5 | **42.0** |
>
> **[Results (Avg@4) of 4B model on Sudoku (Train with L3-L4)]**
>
> |  | atomic action |  |  |  | single execution |  |  |  | macro-action |  |  |  |
> | :---- | :---: | :---: | :---: | :---: | :---: | :---: | :---: | :---: | :---: | :---: | :---: | :---: |
> |  | L1 | L2 | L3 | L4 | L1 | L2 | L3 | L4 | L1 | L2 | L3 | L4 |
> | epoch 1 | 79.8 | 57.5 | 36.8 | 13.3 | **99.8** | **96.0** | **90.0** | **78.0** | 96.3 | 93.5 | 79.5 | 59.0 |
> | epoch 2 | 58.3 | 37.8 | 12.8 | 2.5 | 97.5 | 95.0 | 85.5 | 60.3 | **99.3** | **98.0** | **95.3** | **76.8** |
> | epoch 3 | 2.3 | 0.0 | 0.0 | 0.0 | 85.0 | 65.8 | 40.5 | 18.8 | **99.0** | **98.3** | **93.3** | **79.0** |
> | epoch 4 | \- | \- | \- | \- | 84.8 | 61.3 | 38.0 | 16.3 | **99.5** | **97.5** | **91.3** | **77.0** |
>
> These results are consistent with our core findings, providing direct evidence that the horizon reduction principle generalizes beyond discrete, fully observable puzzle environments.
>
> [1] Webshop: Towards scalable real-world web interaction with grounded language agents.
> [2] RAGEN: Understanding Self-Evolution in LLM Agents via Multi-Turn Reinforcement Learning
> [3] Meta-RL Induces Exploration in Language Agents
> [4] SkillRL: Evolving Agents via Recursive Skill-Augmented Reinforcement Learning
>
> ---
> ## W2. Using a Single RL Algorithm
> As noted in the Appendix D.3, our final algorithm was selected after conducting stability experiments across a range of algorithmic design choices, making it the most stable RL method within our setup. That said, we agree that verifying whether our core empirical findings hold under alternative algorithms is a valuable concern.
>
> To address this, we conduct additional experiments using a GRPO-style method with group normalization, replicating the horizon comparison experiment from Figure 4. The results are presented below. These results are consistent with our original findings, suggesting that the observed training instability is not an artifact of our specific optimization setup but rather a more fundamental consequence of increasing horizon length.
>
> **[Results (Avg@4) of GRPO-style method]**
>
> |  | default |  |  |  | w/ horizon reduction |  |  |  |
> | :---- | :---: | :---: | :---: | :---: | :---: | :---: | :---: | :---: |
> |  | L1 | L2 | L3 | L4 | L1 | L2 | L3 | L4 |
> | epoch 1 | **85.3** | **73.0** | **51.8** | **28.0** | 81.8 | 65.5 | 42.0 | 22.3 |
> | epoch 2 | **97.3** | **90.5** | **78.3** | **57.3** | 94.0 | 86.5 | 78.0 | 52.5 |
> | epoch 3 | 91.0 | 78.5 | 52.0 | 22.5 | **97.5** | **94.0** | **86.0** | **67.3** |
> | epoch 4 | 81.8 | 61.0 | 34.8 | 12.8 | **97.3** | **93.5** | **88.0** | **71.8** |
>
> ---
> ## W3. No comparison with Dense Reward
> We thank the reviewer for this thoughtful suggestion. However, we respectfully contend that treating dense reward signals and PRMs as baselines would not fully align with the analytical focus of this work. Our decision to use verifiable subgoals instead of a learned PRM is intentional. Because the primary objective of this study is to isolate and analyze horizon effects, introducing a learned reward model would introduce an additional confounding factor, making it difficult to attribute observed training dynamics solely to variations in horizon length. In contrast, verifiable subgoals offer a precise and reliable reward signal, ensuring that horizon length remains the only variable under investigation.

---

> > ### Author Rebuttal · Reviewer_PBJi · 2026-04-04
> >
> > Thank the authors for the responses. The additional experiments and explanations are clear. I still remain concerned about how this method will work on more complex environment (like SWE-Gym, tau^2-bench etc.). I remain my score.

---

### Official Review · Reviewer_M8LV · 2026-03-22

**Soundness:** 3
**Presentation:** 3
**Significance:** 2
**Originality:** 3
**Overall Recommendation:** 4
**Confidence:** 3

**Summary:**

This paper presents an in-depth empirical study on the training challenges faced by Large Language Model (LLM) agents in long-horizon tasks. By constructing controlled task environments (such as Sudoku and Rush Hour), the authors successfully decouple 'Goal Distance' from 'Reasoning Complexity,' revealing that the horizon length itself is a critical bottleneck leading to instability and performance collapse in Reinforcement Learning (RL) training. The study proposes the 'Horizon Reduction' principle, which stabilizes RL training and enhances performance through the introduction of macro actions and subgoal decomposition. Furthermore, the paper identifies and defines the phenomenon of 'Horizon Generalization,' demonstrating that models trained on short-horizon tasks can effectively transfer to longer-horizon task variants.

**Compliance With Llm Reviewing Policy:**

Affirmed.

**Final Justification:**

The paper presents a clear and well-executed controlled study on the role of horizon length in RL training for LLM agents. Its main strengths are the careful experimental design, the clarity of presentation, and the practical value of the horizon-reduction insight.

My primary concern was the limited realism of the evaluation environments and whether the conclusions would extend to more complex settings. The rebuttal addresses much of this concern through additional results and theoretical clarification, which strengthen the paper substantially. While I remain somewhat cautious about broader scale-up claims, I now find the core contribution sufficiently supported and useful to the community. Overall, this update moves me to weak accept.

**Key Questions For Authors:**

Please refer to the weaknesses above for details. Overall, my main concern is that the selected environments are overly simplified, which may limit the generalizability of the proposed approach to more complex and realistic settings.

**Limitations:**

yes

**Strengths And Weaknesses:**

Strengths:

1: Innovative Topic Selection. Most current LLM agent studies focus on system optimization or instruction tuning. However, this paper distinguishes itself by keenly identifying and quantifying how horizon length impacts training dynamics. This approach is highly insightful and provides a fresh perspective on the field.

2: Rigorous Experimental Design. By utilizing proxy tasks and professional solvers (such as HoDoKu) to screen tasks, the authors ensured consistent reasoning difficulty across varying horizon lengths. This effectively eliminated confounding factors, making the experimental conclusions significantly more persuasive.

3: Clear and Actionable Insight. The paper distills its findings into a simple and practical design principle—horizon reduction—which is easy to understand and can be readily applied in practice. This makes the contribution valuable beyond the specific experimental setup.

4: Clear Presentation. The paper is clearly written and well-structured.

Weakness:

1: Experimental Limitations. The current study focuses primarily on discrete, symbolic text-based games (e.g., Sudoku). While this approach is effective for controlling variables, the generalizability of the result to Embodied AI, which involves high-dimensional continuous action spaces and complex visual perception, remains to be further validated in the context of long-horizon tasks. Also, the experimental evaluation is conducted on relatively small-scale models (e.g., Qwen3-1.7B).

2: Limited novelty of Results. While the paper presents a well-executed empirical study, the core insights are largely consistent with established understanding in the reinforcement learning literature. The challenges associated with long horizon like exploration difficulty and credit assignment are well-known.

3: Lack of theoretical analysis. The paper primarily focuses on empirical observations and does not provide a formal theoretical framework to explain the impact of horizon length on training dynamics. For instance, there is no analysis of how horizon length affects sample complexity, variance of gradient estimates, or convergence properties.

4: Non-Intrinsic Sequential Structure. The tasks considered in this work do not require interaction in the classical RL sense, as they are fully observable and solvable in a single step. The sequential structure is artificially introduced, which may overstate the role of horizon length as a bottleneck. It remains unclear whether the conclusions extend to genuinely interactive environments(e.g., those with partial observability) where sequential decision-making is unavoidable.

---

> ### Author Rebuttal · Authors · 2026-03-31
>
> ## W1. Experimental Limitations
> We acknowledge that extending our findings to high-dimensional continuous action spaces and visual perception remains an open challenge. The primary goal of this work was to isolate and precisely quantify the effect of horizon length on training dynamics—an analysis that would be confounded in complex real-world environments where goal distance, reasoning complexity, perception difficulty, and action space dimensionality are entangled. Our controlled task design was a deliberate methodological choice. To demonstrate broader applicability, we conduct additional experiments in WebShop [1], a widely adopted benchmark among recent RL for LLM works [2-4], as well as in a POMDP environment. Due to computational constraints, we use Qwen3-1.7B for WebShop; for Sudoku, we additionally report results with a 4B model.
>
> **[Results (Avg@4) on WebShop]**
> |  | default | w/ horizon reduction |
> | :---- | :---- | :---- |
> | epoch 1 | **32.8** | 28.4 |
> | epoch 2 | 27.3 | **31.4** |
> | epoch 3 | 32.8 | **33.6** |
> | epoch 4 | 38.5 | **42.0** |
>
> **[Results (Avg@4) of 4B model on Sudoku (Train with L3-L4)]**
> |  | atomic action |  |  |  | single execution |  |  |  | macro-action |  |  |  |
> | :---: | :---: | :---: | :---: | :---: | :---: | :---: | :---: | :---: | :---: | :---: | :---: | :---: |
> |  | L1 | L2 | L3 | L4 | L1 | L2 | L3 | L4 | L1 | L2 | L3 | L4 |
> | epoch 1 | 79.8 | 57.5 | 36.8 | 13.3 | **99.8** | **96.0** | **90.0** | **78.0** | 96.3 | 93.5 | 79.5 | 59.0 |
> | epoch 2 | 58.3 | 37.8 | 12.8 | 2.5 | 97.5 | 95.0 | 85.5 | 60.3 | **99.3** | **98.0** | **95.3** | **76.8** |
> | epoch 3 | 2.3 | 0.0 | 0.0 | 0.0 | 85.0 | 65.8 | 40.5 | 18.8 | **99.0** | **98.3** | **93.3** | **79.0** |
> | epoch 4 | \- | \- | \- | \- | 84.8 | 61.3 | 38.0 | 16.3 | **99.5** | **97.5** | **91.3** | **77.0** |
>
> These results are consistent with our core findings, providing initial evidence that the horizon reduction principle generalizes beyond our primary experimental setting.
>
> [1] Webshop: Towards scalable real-world web interaction with grounded language agents.
> [2] RAGEN: Understanding Self-Evolution in LLM Agents via Multi-Turn Reinforcement Learning
> [3] Meta-RL Induces Exploration in Language Agents
> [4] SkillRL: Evolving Agents via Recursive Skill-Augmented Reinforcement Learning
>
> ---
> ## W2. Limited novelty of Results
> As the RL-for-LLM community turns its attention to long-horizon tasks, existing discussions have largely focused on algorithmic improvements or reward shaping. Our work takes a complementary but distinct perspective: rather than proposing a new algorithm, we ask what are the inherent failure modes of RL training as horizon length increases, and what systemic design principles can address them. By empirically disentangling horizon length from reasoning complexity, we provide a precise characterization of training instability not previously established in this context. Furthermore, the horizon reduction principle offers a concrete and actionable design axis orthogonal to existing algorithmic approaches.
>
> ---
> ## W3. Lack of theoretical analysis
> Below we provide a concise theoretical clarification on how horizon length affects gradient variance in the REINFORCE estimator.
>
> **(i) Fixed per-step error (exploration-limited regime)**. Assume each step succeeds independently with probability $p = 1 - \\epsilon$, and a reward is obtained only if all steps succeed. The expected return then scales as $p^H$, and the signal-to-noise ratio degrades exponentially:
> $\\frac{\\mathrm{Var}(\\hat{g}\_H)}{|\\nabla J\_H|^2} = (1-\\epsilon)^{-H} - 1 \\approx e^{\\epsilon H}$.
> Achieving a constant SNR therefore requires exponentially growing sample complexity in $H$.
>
> **(ii) Fixed success probability (credit-assignment-limited regime)**. Fixing the probability of an informative terminal signal, the gradient estimator $(R_H - b)\\sum_{t=1}^H \\psi_t $ yields $\\mathrm{Var}(\\hat{g}_H) = O(H^2)$, reflecting quadratic variance growth from delayed credit assignment even when exploration difficulty is held constant.
>
> Both failure modes are mitigated by reducing the effective horizon. Subgoal decomposition reduces variance from $O(H^2)$ to $O(H\\ell)$  for subproblems of length $\\ell \\ll H$. Full derivations will be provided in the appendix.
>
> ---
> ## W4. Non-Intrinsic Sequential Structure
> Our primary object of study is the effective horizon, defined as the number of interaction steps a policy actually requires. Even when a task admits a shorter solution in principle, learning dynamics are governed by the effective decision horizon rather than the task’s inherent structure. The ablation in Fig. 4, which restores a long interaction horizon while holding the macro-action representation fixed, confirms that training collapse is driven by effective horizon length. We additionally conduct experiments in a POMDP environment and observe qualitatively consistent results, to be incorporated into the camera-ready version.

---

> > ### Author Rebuttal · Reviewer_M8LV · 2026-04-05
> >
> > The rebuttal addresses most of my technical concerns, and I think the paper now makes a clearer and better-supported empirical contribution. I still remain somewhat cautious about whether the current environments are sufficient to substantiate stronger claims about scale-up to more realistic settings. Still, given the quality of the controlled study and the practical insight it provides, I am now leaning weak accept.

---

### Decision · Program_Chairs · 2026-04-30

**Decision:**

Accept (regular)

**Comment:**

This paper presents an empirical study examining how task horizon length affects the training dynamics of LLMs used as interactive agents. The authors keep the reasoning complexity to be similar across tasks while change the "goal distance" using environments such as Sudoku and Rush Hour. The main finding of the paper is that increasing the horizon length alone causes severe training instability even if the fundamental "reasoning complexity" of the tasks remains unchanged. Given this observation, the paper argues that we need to perform "horizon reduction" in order to achieve better performance and generalizability.

The main concern shared among the reviewers was the limited scope and realism of the initial evaluation, which relied heavily on discrete, text-based puzzle games with a 1.7B model and a single reinforcement learning algorithm. During the rebuttal, the authors provided supplementary experiments on the partially observable WebShop benchmark, evaluating a larger 4B parameter model, and testing alternative RL optimizers (GRPO), which was able to resolve most concerns.

However, a remaining outstanding problem is the experimental setup. The paper shows that using "macro-actions" leads to better performance compared to "micro-actions". However, this is arguably a strong simplification of the problem as many action sequences cannot be cleanly grouped into meta-actions. This also relates to the concern regarding comparisons with reward shaping or dense reward settings, as reward shaping is rather easy when the action sequences can be cleanly decomposed.